# Improving protein optimization with smoothed fitness landscapes

**Andrew Kirjner**[*]
Massachusetts Institute of Technology
kirjner@mit.edu

**Jason Yim**[*]
Massachusetts Institute of Technology
jyim@mit.edu

**Raman Samusevich**
IOCB Prague, Czech Academy of Sciences,
CIIRC, Czech Technical University in Prague,
University of Chemistry and Technology, Prague
raman.samusevich@uochb.cas.cz

**Shahar Bracha**
Massachusetts Institute of Technology
shaharbr@mit.edu

**Tommi Jaakkola**[†]
Massachusetts Institute of Technology
tommi@csail.mit.edu

**Regina Barzilay**[†]
Massachusetts Institute of Technology
regina@csail.mit.edu

**Ila Fiete**[†]
Massachusetts Institute of Technology
fiete@mit.edu

## Abstract

The ability to engineer novel proteins with higher fitness for a desired property would be revolutionary for biotechnology and medicine. Modeling the combinatorially large space of sequences is infeasible; prior methods often constrain optimization to a small mutational radius, but this drastically limits the design space. Instead of heuristics, we propose *smoothing* the fitness landscape to facilitate protein optimization. First, we formulate protein fitness as a graph signal then use Tikunov regularization to smooth the fitness landscape. We find optimizing in this smoothed landscape leads to improved performance across multiple methods in the GFP and AAV benchmarks. Second, we achieve state-of-the-art results utilizing discrete energy-based models and MCMC in the smoothed landscape. Our method, called Gibbs sampling with Graph-based Smoothing (GGS), demonstrates a unique ability to achieve 2.5 fold fitness improvement (with *in-silico* evaluation) over its training set. GGS demonstrates potential to optimize proteins in the limited data regime. Code: https://github.com/kirjner/GGS

## 1 Introduction

In protein engineering, fitness can be defined as performance on a desired property or function. Examples of fitness include catalytic activity for enzymes (Anderson et al., 2021) and fluorescence for biomarkers (Remington, 2011). Protein optimization seeks to improve protein fitness by altering the underlying sequences of amino acids. However, the number of possible proteins increases exponentially with sequence length, rendering it infeasible to perform brute-force search to engineer novel functions, which often require multiple mutations from the starting sequence (i.e. at least 3 (Ghafari & Weissman, 2019)). Directed evolution (Arnold, 1998) has been successful in improving protein fitness, but it requires substantial labor and time.

---

[*]Contributed equally to this work. Authors agreed ordering can be changed for their respective interests.
[†]Advised equally to this work.

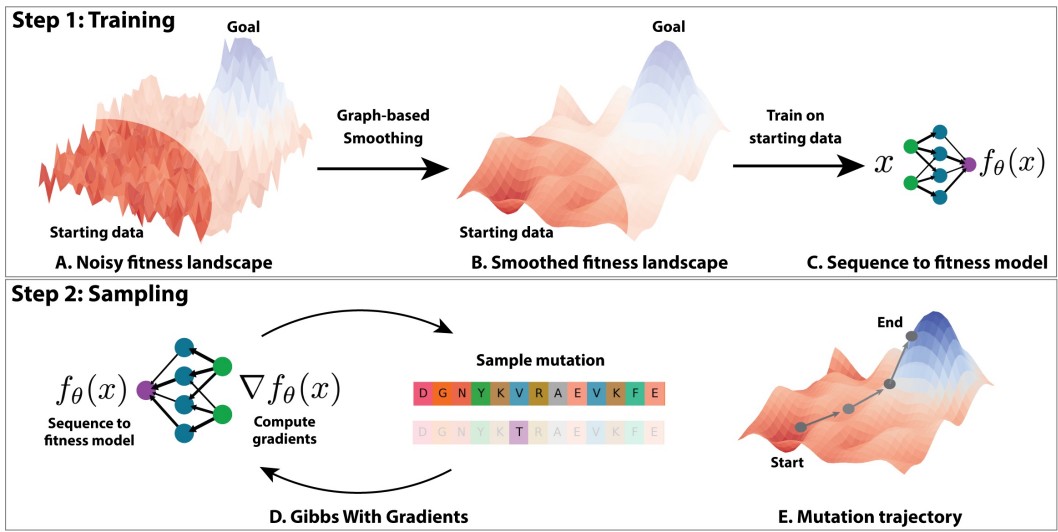

Figure 1: Overview. **(A)** Protein optimization is challenging due to a noisy fitness landscape where the starting dataset (unblurred) is a fraction of the landscape with the highest fitness sequences hidden (blurred). **(B)** We develop Graph-based Smoothing (GS) to estimate a smoothed fitness landscape from the starting data. **(C)** A model is trained on the smoothed fitness landscape to infer the rest of the landscape. **(D)** Gradients from the model are used in Gibbs With Gradients (GWG) where on each step a new mutation is proposed. **(E)** The goal of sampling is for each trajectory to gradually head towards higher fitness.

We aim to computationally generate high-fitness proteins by optimizing a learned model of the fitness landscape, but face several challenges. Proteins can be notorious for highly non-smooth fitness landscapes[1]: fitness can change dramatically with single mutations, fitness measurements contain experimental noise, and most protein sequences have zero fitness (Brookes et al., 2022). Furthermore, protein fitness datasets are scarce and difficult to generate due to their high costs (Dallago et al., 2021). As a result, machine learning (ML) methods are susceptible to predicting false positives and getting stuck in local optima (Brookes et al., 2019). The 3D protein structure, if available, can provide information in navigating the noisy fitness landscape such as identifying hot spot residues (Zerbe et al., 2012), but high quality structures are not available in many cases.

One way to deal with noisy and limited data is to *regularize* the fitness landscape model[2]. Our work considers a *smoothing* regularizer in which similar sequences (based on a distance measure) are predicted to have similar predicted fitness. While actual fitness lanscapes are not smooth, smoothing can be an important tool in the context of optimization, allowing gradient-based methods to reach higher peaks by avoiding local optima, especially in discrete optimization (Zanella, 2020). A few works have studied properties of protein fitness landscapes (Section 2), but none have directly applied smoothing with a graph framework during optimization.

We propose a novel method for applying smoothing to protein sequence and fitness data together with an optimization technique that takes advantage of the smoothing. First, we formulate sequences as a graph with fitness values as node attributes and apply Tikunov regularization to smooth the topological signal measured by the graph Laplacian. The smoothed data is then fitted with a neural network to be used as a model for discrete optimization (Figure 1 top). Second, we sample over the energy function for high fitness sequences by using the model's gradients in a Gibbs With Gradients (GWG) procedure (Grathwohl et al., 2021). In GWG, a discrete distribution is constructed based on the model's gradients where mutations with improved fitness will correlate with higher probability. The process of taking gradients and sampling mutations is performed in an iterative fashion where subsequent mutations will guide towards higher fitness (Figure 1 bottom).

---

[1]Landscape refers to the sequence to fitness mapping.
[2]In the sequel, we will use "model" when referring to the fitness landscape model.

Figure 1 shows an overview of the method. We refer to the procedure of smoothing then sampling as Gibbs sampling with Graph-based Smoothing (GGS). To evaluate our method, we introduce a set of tasks using the well studied Green Fluorescent Proteins (GFP) (Sarkisyan et al., 2016) and Adeno-Associated Virus (AAV) (Bryant et al., 2021) proteins. We chose GFP and AAV because of their real-world importance and availability of large mutational data. We design a set of tasks that emulate starting with noisy and limited data and evaluate with a trained model (as done in most prior works). We evaluate GGS and prior works on our proposed benchmarks to show that GGS is state-of-the-art in GFP and AAV fitness optimization. Our contributions are summarized as follows:

- We develop a novel sequence-based protein optimization algorithm, GGS, which uses graph-based smoothing to train a smoothed fitness model. The model is used as a discrete energy function to progressively sample mutations towards higher-fitness sequences with GWG (Section 3).
- We develop a set of tasks that measure a method's ability to extrapolate towards higher fitness. We use publicly available GFP and AAV datasets to emulate difficult optimization scenarios of starting with limited and noisy data (Section 4.1).
- Our benchmark shows prior methods fail to extrapolate towards higher fitness. However, we show graph-based smoothing can drastically improve their performance; in one baseline, the fitness jumps from 18% to 39% in GFP and 4% to 44% in AAV after smoothing (Section 4.2).
- Our method GGS directly exploits smoothness to achieve state-of-the-art results with 5 times higher fitness in GFP and 2 times higher in AAV compared to the next best method (Section 4.2).

## 2 RELATED WORK

**Protein optimization and design.** Approaches can broadly be categorized using sequence, structure or both. Sequence-based methods have been explored through the lens of reinforcement learning (Angermueller et al., 2020), latent space optimization (Stanton et al., 2022; Lee et al.; Maus et al., 2022), generative models (Notin et al., 2022; Meier et al., 2021; Jain et al., 2022; Gruver et al., 2023), and model-based directed evolution (Sinai et al., 2020; Padmakumar et al., 2023; Ren et al., 2022). Together they face the issue of a noisy fitness landscape to optimize. We focus on sequence-based methods using Gibbs With Gradients (GWG) (Grathwohl et al., 2021) which can perform state-of-the-art in discrete optimization but requires a smooth energy function for strong performance. Concurrently, Emami et al. (2023) used GWG for protein optimization with a product of experts distribution using a protein language model. However, they achieved subpar results.

Previous methods focused on developing new sampling and optimization techniques. Our work is complimentary by addressing the need for improved regularization and smoothing. *We show in our experiments that our smoothing technique can enhance the performance of prior methods.*

**Protein fitness regularization.** The NK model was an early attempt to model smoothness of protein fitness through a statistical model of epistasis (Kauffman & Weinberger, 1989). Brookes et al. (2022) proposed a framework to approximate the sparsity of protein fitness using a generalized NK model (Buzas & Dinitz, 2013). Concurrently, dWJS (Frey et al., 2023) is most related to our work by utilizing Gaussian noise to regularize the discrete energy function during Langevin MCMC. dWJS trains by denoising to smooth a energy-based model whereas we apply discrete regularization using graph-based smoothing techniques.

Finally, we distinguish our smoothing method from traditional regularizers applied during training such as dropout (Srivastava et al., 2014). Our goal is to smooth the fitness landscape in a way that is amenable for iterative optimization. We enforce similar sequences to have similar fitness which is not guaranteed with dropout or similar regularizers applied in minibatch training. *Evaluating multiple smoothing strategies is not the focus of our work, but rather to demonstrate their importance.*

## 3 METHOD

The following describes our method. Section 3.1 details the problem formulation. Next section 3.2 describes the procedure for training a smoothed model. Lastly, section 3.3 provides background on Gibbs With Gradients (GWG) which is adapted for protein optimization. The full algorithm, Gibbs sampling with Graph-based Smoothing (GGS), is presented in Algorithm 1.

## 3.1 PROBLEM FORMULATION

We denote the starting set of $N$ proteins as $\mathcal{D} = (X, Y)$ where $X = \{x_1, \ldots, x_N\} \subset \mathcal{V}^M$ are the sequences and $Y = \{y_1, \ldots, y_N\}$ are corresponding real-valued scalar fitness measurements. Each sequence $x_i \in \mathcal{V}^M$ is composed of $M$ residues from a vocabulary $\mathcal{V}$ of 20 amino acids. Subscripts refer to different sequences. Note our method can be extended to other modalities, e.g. nucleic acids.

For *in-silico* evaluation, we denote the set of all known sequences and fitness measurements as $\mathcal{D}^* = (X^*, Y^*)$. We assume there exists a unknown black-box function $g : \mathcal{V}^M \to \mathbb{R}$ such that $g(x^*) = y^*$. In practice, $g$ needs to be approximated by a evaluator model, $g_\phi$, trained with weights $\phi$ to minimize prediction error on $\mathcal{D}^*$. $g_\phi$ poses a limitation to evaluation since the true fitness needs to be verified with biological experiments. Nevertheless, an *in-silico* approximation provides a accessible way for evaluation and is done in all prior works. The starting dataset is a strict subset of the known dataset $\mathcal{D} \subset \mathcal{D}^*$ to simulate fitness optimization scenarios. Given $\mathcal{D}$, our task is to generate a set of sequences with higher fitness than the starting set.

## 3.2 GRAPH-BASED SMOOTHING ON PROTEINS

Our goal is to develop a model of the sequence-to-fitness mapping that can be utilized when sampling higher fitness sequences. Unfortunately, the high-dimensional sequence space coupled with few data points and noisy labels can result in a noisy model that is prone to sampling false positives or getting stuck in local optima. To address this, we use smoothing techniques from graph signal processing.

The smoothing process is depicted in Figure 2. First, we train a noisy fitness model $f_{\tilde{\theta}} : \mathcal{V}^M \to \mathbb{R}$ with weights $\tilde{\theta}$ on the initial dataset $\mathcal{D}$ using Mean-Squared Error (MSE). $\mathcal{D}$ is usually very small in real-world scenarios. We augment the dataset by using $f_{\tilde{\theta}}$ to infer the fitness of neighboring sequences which we do not have labels for – known as transductive inference. Neighboring sequences are generated by randomly applying point mutations to each sequence in $X$. The augmented and original sequences become nodes, $V$, in our graph while their fitness labels are node attributes. Edges, $\mathcal{E}$, are constructed with a $k$-nearest neighbor (kNN) graph around each node based on the Levenshtein distance[3]. The graph construction algorithm can be found in Algorithm 4.

The following borrows techniques from Isufi et al. (2022). The smoothness of the fitness variability in our protein graph is defined as the sum over the square of all local variability,

$$\mathsf{TV}_2(Y) = \frac{1}{2} \sum_{i \in \mathcal{V}} (\Delta y_i)^2, \quad \Delta y_i = \sqrt{\sum_{(i,j) \in \mathcal{E}} (y_i - y_j)^2}.$$

TV refers to Total Variation and $\Delta y_i$ is the local variability of node $i$ that measures local changes in fitness. Using $\mathsf{TV}_2$ as a regularizer, we solve the following optimization problem, known as Tikhunov regularization (Zhou & Schölkopf, 2004), for a new set of smoothed fitness labels,

$$\underset{\hat{Y} \in \mathbb{R}^{|V|}}{\arg\min} \|Y - \hat{Y}\|_2^2 + \gamma \, \mathsf{TV}_2(\hat{Y}). \tag{1}$$

With abuse of notation, we represent $Y$ as a vector with each node's fitness. $\gamma$ is a hyperparameter set to control the smoothness; too high can lead to underfitting. We experiment with different $\gamma$'s in Section 4. Since eq. (1) is a quadratic convex problem, it has a closed form solution, $\hat{Y} = (\mathbb{I} + \gamma L)^{-1} Y$ where $L$ is the graph Laplacian and $\mathbb{I}$ is the identity matrix. The final step is to retrain the model on the sequences in the graph and their smoothed fitness labels. The result will be a model $f_\theta$ with lower $\mathsf{TV}_2$ than before and thus improved smoothness. The smoothing algorithm is in Algorithm 2.

## 3.3 SAMPLING IMPROVED FITNESS WITH GIBBS

Equipped with model $f_\theta$ from section 3.2, we apply it in a procedure to sample mutations that improve the starting sequences' fitness. $f_\theta$ can also be viewed as an energy-based model (EBM) that defines a Boltzmann distribution $\log p(x) = f_\theta(x) - \log Z$ where $Z$ is the normalization constant. Higher fitness sequences will be more likely under this distribution, while sampling will induce diversity and novelty. To sample from $p(x)$, we use Gibbs With Gradients (GWG) Grathwohl et al. (2021) which

---

[3]Defined as the minimum number of mutations between two sequences.

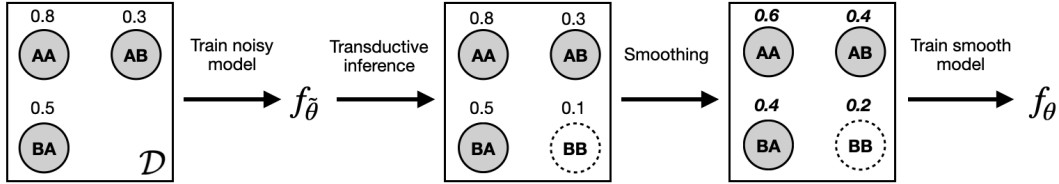

Figure 2: Steps in graph-based smoothing on proteins illustrated with a fictitious data of length 2 sequences with vocabulary $\{A, B\}$. Above each node are corresponding fitness values. Solid nodes are those in our training set while dashed nodes are augmented via point mutations to increase the smoothing effectiveness. See section 3.2 for description of each step.

has attracted significant interest due to its simplicity and state-of-the-art performance in discrete optimization. In this section, we describe the GWG procedure for protein sequences. GWG uses Gibbs sampling with approximations of *locally informed proposals* (Zanella, 2020):

$$q(x'|x) \propto \exp\left(\frac{1}{2}\sum_i (x_i')^\top d_\theta(x)_i\right) \mathbb{1}(x' \in H(x)), \quad d_\theta(x)_i = [\nabla_x f_\theta(x)]_i - x_i \odot [\nabla f_\theta(x)]_i .$$
(2)

With slight abuse of notation, we use the one-hot sequence representation $x \in \{0,1\}^{M \times |\mathcal{V}|}$ where $x_i \in \{0,1\}^{|\mathcal{V}|}$ represents the $i$th index of the sequence with 1 at its amino acid index and 0 elsewhere. $\odot$ is the element wise product. $H(x) = \{y \in \mathcal{V}^M : d_{\text{Hamming}}(x, y) \leq 1\}$ is the 1-ball around $x$ using Hamming distance. The core idea of GWG is to use $d_\theta(x)_i$ as the first order approximation of a *continuous* gradient of the change in likelihood from mutating the $i$th index of $x$ to a different amino acid. The quality of the proposals in eq. (2) rely on the *smoothness* of the energy $f_\theta$ (Theorem 1 in Grathwohl et al. (2021)). If the gradients, $\nabla f_\theta$, are noisy, then the proposal distributions are ineffective in sampling better sequences. Hence, smoothing $f_\theta$ is desirable (see section 4).

The choice of $H(\cdot)$ as the 1-Hamming ball limits $x'$ to *point mutations* from $x$ and only requires $\mathcal{O}(M \times |\mathcal{V}|)$ compute to construct. Let the point mutation where $x$ and $x'$ differ be defined by the residue location, $i^{\text{loc}} \in \{1, \ldots, M\}$, and amino acid substitution, $j^{\text{sub}} \in \{1, \ldots, |\mathcal{V}|\}$. By limiting $x'$ to point mutants $(i^{\text{loc}}, j^{\text{sub}})$, sampling $q(x'|x)$ is equivalent to sampling the following,

$$(i^{\text{loc}}, j^{\text{sub}}) \sim q(\cdot|x) = \text{Cat}\left(\text{Softmax}\left(\left\{\frac{d_\theta(x)_{i,j}}{\tau}\right\}_{i=1,j=1}^{M,|\mathcal{V}|}\right)\right)$$
(3)

where $\tau$ is the sampling temperature and $d_\theta(x)_{i,j}$ is the logits of mutating to $(i, j)$. The proposal sequence $x'$ is constructed by setting its $i^{\text{loc}}$ residue to $j^{\text{sub}}$ and equal to $x$ elsewhere. Each proposed sequence is accepted or rejected using Metropolis-Hasting (MH),

$$\min\left(\exp(f_\theta(x') - f_\theta(x))\frac{q(x|x')}{q(x'|x)}, \; 1\right).$$
(4)

We provide the GWG algorithm in Algorithm 3.

**Clustered sampling.** GWG requires a starting sequence to start mutating. A reasonable starting set are the sequences $X$ used to train the model. On each round $r$, we use eq. (3) to propose $N_{\text{prop}}$ mutations for each sequence. If accepted via eq. (4), then the mutated sequence will be added to the next round. However, this procedure can lead to an intractable number of sequences to consider.

To control compute bandwidth, we perform hierarchical clustering (Müllner, 2011) on all the sequences in a round and take the sequence of each cluster with the highest predicted fitness using $f_\theta$. Let $\mathcal{C}$ be the number of clusters which we set based on amount of available compute. This procedure, known as `Reduce`, is,

$$\text{Reduce}(X; \theta) = \bigcup_{c=1}^{\mathcal{C}} \{\arg\max_{x \in X^c} f_\theta(x)\} \text{ where } \{X^c\}_{c=1}^{\mathcal{C}} = \text{Cluster}(X; \mathcal{C}).$$
(5)

Each round $r$ reduces the sequences from the previous round and performs GWG sampling.

$$\tilde{X}_r = \text{Reduce}(X_r; \theta), \quad X_{r+1} = \text{GWG}(\tilde{X}_r; \theta)$$

To summarize, we adapted GWG for protein optimization by developing a smoothed model to satisfy GWG's smoothness assumptions and use clustering during sampling to reduce redundancy and compute. An illustration of clustered sampling is provided in Figure 5.

The full algorithm for smoothing and clustered sampling is provided in Algorithm 1.

---

**Algorithm 1** GGS: Gibbs sampling with Graph-based Smoothing

---

**Require:** Starting dataset: $\mathcal{D} = (X, Y)$
1: $\tilde{\theta} \leftarrow \arg\max_{\tilde{\theta}} \mathbb{E}_{(x,y)\sim\mathcal{D}} \left[ (y - f_{\tilde{\theta}}(x))^2 \right]$          ▷ Initial training
2: $\theta \leftarrow \text{Smooth}(\mathcal{D}; \tilde{\theta})$          ▷ GS algorithm 2
3: **for** $r = 0, \ldots, R - 1$ **do**
4:     $\tilde{X}_r \leftarrow \text{Reduce}(X_r; \theta)$
5:     $X_{r+1} \leftarrow \text{GWG}(\tilde{X}_r; \theta)$          ▷ GWG algorithm 3
6: **end for**
7: **Return** $\text{TopK}(X_R)$      ▷ Return Top-K best sequences based on predicted fitness $f_\theta$

---

## 4 EXPERIMENTS

Our experiments demonstrate the benefits of smoothing in protein optimization. Section 4.1 presents a set of challenging tasks based on the GFP and AAV proteins that emulate starting with experimental noise and a sparsely sampled fitness landscape. Section 4.2 evaluates the performance of baselines and our method, GGS, on our benchmark. In addition, we find applying smoothing improves performance for two baselines. Section 4.3 provides sweeps over hyperparameters and analysis of GGS.

**Baselines.** We choose a representative set of prior works that evaluated on GFP and AAV: GFlowNets (GFN-AL) (Jain et al., 2022), model-based adaptive sampling (CbAS) (Brookes et al., 2019), greedy search (AdaLead) (Sinai et al., 2020), bayesian optimization (BO-qei) (Wilson et al., 2017), conservative model-based optimization (CoMs) (Trabucco et al., 2021), and proximal exploration (PEX) (Ren et al., 2022). NOS (Gruver et al., 2023) performs protein optimization with diffusion models. However, their framework is tailored to antibody optimization and requires non-trivial modifications for general proteins. We were unable to evaluate Song & Li (2023) due to unrunnable public code.

**GGS implementation.** We use a 1D CNN (see Appendix B.1 for architecture and training) for model $f_\theta$. To ensure a fair comparison, we use the same model architecture in baselines when possible. In graph-based smoothing (GS), we augment the graph until it has $N_{\text{nodes}} = 250,000$ nodes. We found larger graphs to not give improvements. Similarly, we use $\tau = 0.1$, $R = 15$ rounds and $N_{\text{prop}} = 100$ proposals per round during GWG at which sequences would converge and more sampling did not give improvements. We choose the smoothing weight $\gamma = 1.0$ through grid search. We study sensitivity to hyperparameters, especially $\gamma$, in Section 4.3.

### 4.1 BENCHMARK

We develop a set of tasks based on two well-studied protein systems: Green Fluoresent Protein (GFP) and Adeno-Associated Virus (AAV) (Sarkisyan et al., 2016; Bryant et al., 2021). These were chosen due to their relatively large amount of measurements, 56,806 and 44,156 respectively, with sequence variability of up to 15 mutations from the wild-type. Other datasets are either too small or do not have enough sequence variability. GFP's fitness is its fluorescence properties as a biomarker while for AAV's is the ability to package a DNA payload, i.e. for gene delivery. We found GFP and AAV to suffice in demonstrating how prior methods fail to extrapolate.

One measure of difficulty is the number of mutations required to achieve the highest known fitness; this assesses a method's exploration capability. We designate the set of optimal proteins, $X^{99\text{th}}$, as any sequence in the 99th fitness percentile in the entire dataset[4]. Quantitatively, we compute the *minimum* number of mutations required from the training set to achieve the optimal fitness:

$$\text{Gap}(X_0; X^{99\text{th}}) = \min(\{\text{dist}(x, \tilde{x}) : x \in X, \tilde{x} \in X^{99\text{th}}\}). \tag{6}$$

---

[4]This may differ from the true optimal protein found in nature. Unfortunately, we must work with existing datasets since every possible protein cannot be experimentally measured.

A high mutational gap would require the method discovering many mutations in a high dimensional space. A second measure of difficulty is the fitness range of the starting set of sequences. Starting with a low range of fitness requires the method to learn from barely functional proteins and exploit limited knowledge to find mutations that confer higher fitness. Appendix A shows Gap and starting rate are necessary as we found the previous GFP benchmark (Trabucco et al., 2022) as too "easy" by only requiring one mutation to achieve the optimal fitness.

Recall the protein optimization task is to use the starting set $\mathcal{D}$ to propose a set of sequences with higher fitness. We design two difficulties, *medium* and *hard*, for GFP and AAV based on the properties of $\mathcal{D}$. We restricted the range and the mutational gap to modulate task difficulty. We found Gap= 7 and Range $< 30\%$ to suffice in finding where our baseline methods fail to discover better proteins. We use this setting as the hard difficulty and sought to develop GGS to solve it.

<table>
<tr><td colspan="4" align="center">Table 1: GFP tasks</td><td colspan="4" align="center">Table 2: AAV tasks</td></tr>
<tr><td>Difficulty</td><td>Range (%)</td><td>Gap</td><td>$|\mathcal{D}|$</td><td>Difficulty</td><td>Range (%)</td><td>Gap</td><td>$|\mathcal{D}|$</td></tr>
<tr><td>Medium</td><td>20th-40th</td><td>6</td><td>2828</td><td>Medium</td><td>20th-40th</td><td>6</td><td>2139</td></tr>
<tr><td>Hard</td><td>$< 30$th</td><td>7</td><td>2426</td><td>Hard</td><td>$< 30$th</td><td>7</td><td>3448</td></tr>
</table>

***In-silico* evaluation.** We follow prior works in using a trained evaluator model as a proxy for real-world experimental validation. A popular model choice is the TAPE transformer (Rao et al., 2019). However, we noticed a poor performance of the transformer compared to a simpler CNN that matches the findings of Dallago et al. (2021). We use CNN architecture for the evaluator due to its superior performance. Following Jain et al. (2022), each method generates 128 samples $\hat{X} = \{\hat{x}_i\}_{i=1}^{128}$ whose approximated fitness is predicted with the evaluator. We additionally report Diversity and Novelty that are also used in Jain et al. (2022). Descriptions of these metrics can be found in Appendix B.2 We emphasize that higher diversity and novelty are *not* equivalent to better performance, but provide insight into the exploration and exploitation trade-offs of different methods. For instance, a random algorithm would achieve maximum diversity and novelty.

## 4.2 RESULTS

We run 5 seeds and report the average metric across all seeds including the standard deviation in parentheses. We evaluate GGS and previously described baselines. To ensure a fair comparison, we use the same CNN architecture as the model across all methods – all our baselines (and GGS) perform model-based optimization. Since graph-based smoothing (GS) is a general technique, we sought to evaluate its effectiveness in each of our baselines. To incorporate GS, we used the smoothed predictor as a replacement in each baseline which will be denoted with "+ GS". Table 3 summarizes GFP results while table 4 summarizes AAV.

GGS substantially outperforms all unsmoothed baselines, consistently achieving a improvement in fitness from the starting range of fitness in each difficulty. However, the smoothed baselines (lines with + GS) demonstrated a up to three fold improvement for CbAS, AdaLead. We find larger improvements in GFP where the sequence space is far larger than AAV – suggesting the GFP fitness landscape is harder to optimize over.

The most difficult task is clearly hard difficulty on GFP where all the baselines without smoothing cannot achieve fitness higher than the training set. With smoothing, GGS achieves the best fitness since the sampling procedure uses gradient-based proposals that benefit from a smooth model. Appendix C.2.1 presents results on additional difficulties to analyze GGS beyond hard..

We observe GGS is able to achieve the highest fitness while exhibiting respectable diversity and novelty. Notably, GGS's novelty falls within the range of the mutational gap in each difficulty, suggesting it is extrapolating an appropriate amount for each task. Our sampling procedure, GWG, fails to perform without smoothing which agrees with its theoretical requirements of requiring a smooth model for good performance. We conclude smoothing is a beneficial technique not only for GGS but also for some baselines. GGS is able to achieve state-of-the-art results in our benchmark.

Table 3: GFP optimization results. Bold indicates improvement with smoothing.

| Method | Medium difficulty | | | Hard difficulty | | |
|---|---|---|---|---|---|---|
| | Fitness | Diversity | Novelty | Fitness | Diversity | Novelty |
| GFN-AL | 0.09 (0.1) | 25.1 (0.5) | 213 (2.2) | 0.1 (0.2) | 23.6 (1.0) | 214 (4.2) |
| GFN-AL + GS | **0.15 (0.1)** | 16.3 (1.6) | 213 (2.7) | **0.16 (0.2)** | 22.2 (0.8) | 215 (4.6) |
| CbAS | 0.14 (0.0) | 9.7 (1.1) | 7.2 (0.4) | 0.18 (0.0) | 9.6 (1.3) | 7.8 (0.4) |
| CbAS + GS | **0.66 (0.1)** | 3.8 (0.4) | 5.0 (0.0) | **0.57 (0.0)** | 4.2 (0.17) | 6.3 (0.6) |
| AdaLead | 0.56 (0.0) | 3.5 (0.1) | 2.0 (0.0) | 0.18 (0.0) | 5.6 (0.5) | 2.8 (0.4) |
| AdaLead + GS | **0.59 (0.0)** | 5.5 (0.3) | 2.0 (0.0) | **0.39 (0.0)** | 3.5 (0.1) | 2.0 (0.0) |
| BOqei | 0.20 (0.0) | 19.3 (0.0) | 0.0 (0.0) | 0.0 (0.5) | 94.6 (71) | 54.1 (81) |
| BOqei + GS | 0.08 (0.0) | 19.3 (0.0) | 0.0 (0.0) | 0.01 (0.0) | 13.4 (0.0) | 0.0 (0.0) |
| CoMS | 0.0 (0.1) | 133 (25) | 192 (12) | 0.0 (0.1) | 144 (7.5) | 201 (3.0) |
| CoMS + GS | 0.0 (0.5) | 129 (25) | 128 (84) | 0.0 (0.1) | 114 (36) | 187 (5.7) |
| PEX | 0.47 (0.0) | 3.0 (0.0) | 1.4 (0.2) | 0.0 (0.0) | 3.0 (0.0 | 1.3 (0.3) |
| PEX + GS | 0.45 (0.0) | 2.9 (0.0) | 1.2 (0.3) | 0.0 (0.0) | 2.9 (0.0) | 1.2 (0.3) |
| GWG | 0.1 (0.0) | 33.0 (0.8) | 12.8 (0.4) | 0.0 (0.0) | 4.2 (7.0) | 7.6 (1.1) |
| **GGS** (ours) | **0.76 (0.0)** | 3.7 (0.2) | 5.0 (0.0) | **0.74 (0.0)** | 3.6 (0.1) | 8.0 (0.0) |

Table 4: AAV optimization results. Bold indicates improvement with smoothing.

| Method | Medium difficulty | | | Hard difficulty | | |
|---|---|---|---|---|---|---|
| | Fitness | Diversity | Novelty | Fitness | Diversity | Novelty |
| GFN-AL | 0.2 (0.1) | 9.6 (1.2) | 19.4 (1.1) | 0.1 (0.1) | 11.6 (1.4) | 19.6 (1.1) |
| GFN-AL + GS | 0.18 (0.1) | 9.0 (1.1) | 20.6 (0.5) | 0.1 (0.1) | 9.5 (2.5) | 19.4 (1.1) |
| CbAS | 0.43 (0.0) | 12.7 (0.7) | 7.2 (0.4) | 0.36 (0.0) | 14.4 (0.7) | 8.6 (0.5) |
| CbAS + GS | **0.47 (0.1)** | 8.8 (0.9) | 5.3 (0.6) | **0.4 (0.0)** | 12.5 (0.4) | 7.0 (0.0) |
| AdaLead | 0.46 (0.0) | 8.5 (0.8) | 2.8 (0.4) | 0.4 (0.0) | 8.53 (0.1) | 3.4 (0.5) |
| AdaLead + GS | 0.43 (0.0) | 3.77 (0.2) | 2.0 (0.0) | **0.44 (0.0)** | 2.9 (0.1) | 2.0 (0.0) |
| BOqei | 0.38 (0.0) | 15.22 (0.8) | 0.0 (0.0) | 0.32 (0.0) | 17.9 (0.3) | 0.0 (0.0) |
| BOqei + GS | 0.34 (0.0) | 12.2 (0.3) | 0.0 (0.0) | 0.32 (0.0) | 17.2 (0.7) | 0.0 (0.0) |
| CoMS | 0.37 (0.1) | 10.1 (5.9) | 8.2 (3.5) | 0.26 (0.0) | 10.7 (3.5) | 10.0 (2.8) |
| CoMS + GS | 0.37 (0.1) | 9.0 (3.6) | 8.6 (3.7) | 0.22 (0.1) | 13.2 (1.9) | 12.6 (2.4) |
| PEX | 0.4 (0.0) | 2.8 (0.0) | 1.4 (0.2) | 0.3 (0.0) | 2.8 (0.0) | 1.3 (0.3) |
| PEX + GS | 0.4 (0.0) | 2.8 (0.0) | 1.4 (0.2) | 0.3 (0.0) | 2.8 (0.0) | 1.1 (0.2) |
| GWG | 0.43 (0.1) | 6.6 (6.3) | 7.7 (0.8) | 0.33 (0.0) | 12.0 (0.4) | 12.2 (0.4) |
| **GGS** (ours) | **0.51 (0.0)** | 4.0 (0.2) | 5.4 (0.5) | **0.60 (0.0)** | 4.5 (0.5) | 7.0 (0.0) |

## 4.3 ANALYSIS

We analyze the effect of varying the following hyperparameters: number of nodes $N_{\text{nodes}}$ in the protein graph, smoothness weight $\gamma$ in eq. (1), and number of sampling rounds $R$ during GWG sampling. For space, we leave the analysis of the sampling temperature $\tau$ in appendix C.1. Figure 3 presents the results of running GGS with different hyperparameters on the hard difficulty of GFP and AAV. Along the X-axis, we plot the median performance of the sequences during each round of GWG where $r = 0$ is initialization and $r = 15$ are the sequences and the end of GWG. The Y-axis shows the predicted fitness of the smoothed model in blue while the fitness scored with our is shown in red. Interestingly, we find in the majority of cases the smoothed model's predictions are highly correlated

with the evaluator along the sampling trajectory. This is despite the model being trained on 4% of the data with the hard filtering. Appendix C.2.2 shows the prediction error where we find smoothing greatly improves in predicting the fitness of unseen sequences despite having higher train error.

**Graph size.** We find $N_{\text{nodes}} = 250,000$ nodes to have the best performance over a smaller graph with 100,000 nodes. Larger graphs allow for better approximation of the fitness landscape. However, larger graphs require more compute. A future direction could be to determine optimal graph size with different node augmentations strategies than random mutations.

**Smoothing.** Too much smoothing $\gamma = 10.0$ can lead to worse performance in AAV while GFP is not sensitive. This suggests the optimal $\gamma$ is dependent on the particular fitness landscape. Since real proteins landscapes are unknown, the biggest limitation of our method is determining the optimal $\gamma$. An important extension of GGS is to theoretically characterize landscapes (Buzas & Dinitz, 2013) and provide guidelines of selecting $\gamma$.

**Sampling convergence.** We find a set number of rounds are required for GWG sampling to converge when the landscape is smooth enough (middle and right column). We find additional rounds are unnecessary; in practice, more rounds can be ran to ensure convergence. Results on sweeping the temperature are in Appendix C.1 where we see 0.1 clearly performs the best for GFP and AAV.

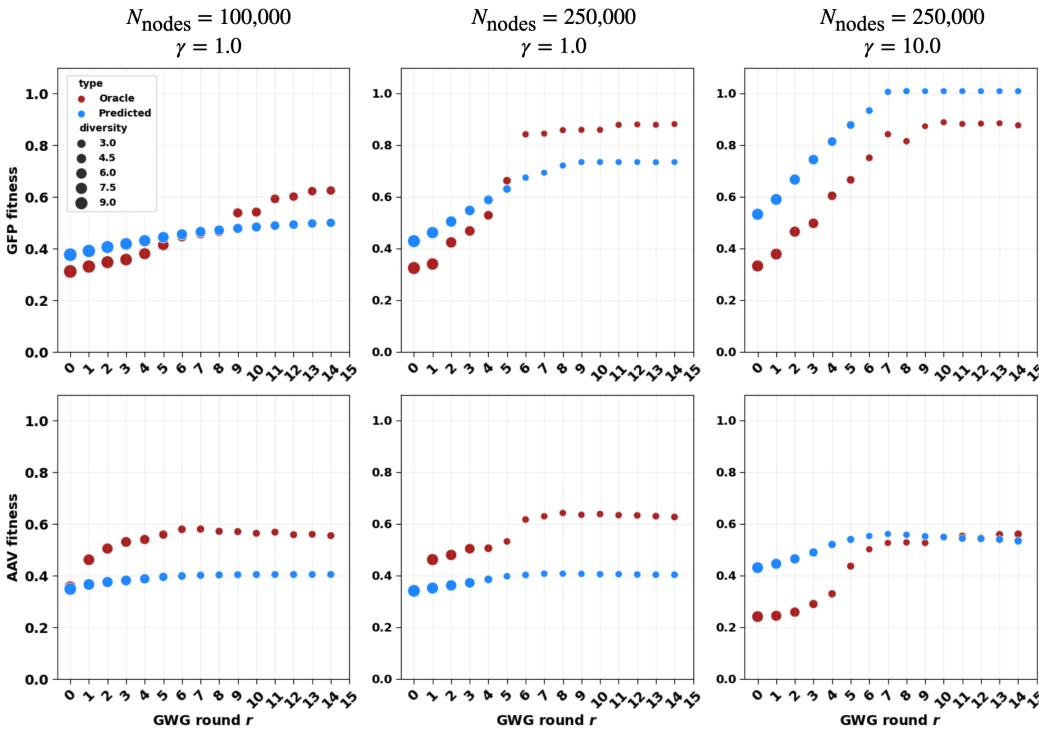

Figure 3: GGS hyperparameter analysis on GFP and AAV hard difficulty. See Section 4.3.

## 5 DISCUSSION

We present Gibbs sampling with Graph-based Smoothing (GGS) for protein optimization with a smoothed fitness landscape. Our main contribution and insight is a novel application of graph signal processing to protein optimization. We show smoothing is not only beneficial to our method but also to our baselines. To evaluate, we designed a suite of tasks around two measure of difficulty: number of edits to achieve the 99th percentile (mutational gap) and starting range of fitness. All baselines struggled to achieve good performance on our tasks. However, some baselines showed a three fold improvement with smoothing. GGS performed the best by combining Gibbs with gradients

with a smoothed model – demonstrating the synergy of gradient-based sampling with a smooth discrete energy-based model. Our results highlight the benefits of optimizing over a smooth landscape that may not be reflective of the true fitness landscape. We believe it's important to investigate how regularization can be used to transform protein fitness data to be compatible with modern optimization algorithms. Our goal is to not learn the excess biological noise, but find the signal in the data to discover the best protein. We conclude with limitations.

**Evaluation limitations.**   The results demonstrate strong evidence of using smoothing given its improvement in multiple methods. Despite this, our evaluations follow prior works by utilizing an trained model for evaluation. This can be unreliable compared to testing out sequences with wet-lab validation. Unfortunately, wet-lab validation can be cost and time intensive. The ultimate test would be to use GGS in an active learning or experimental pipeline with wet-lab validation in the loop.

**Method limitations.**   Our method utilizes several hyperparameters such as the graph size and smoothing parameter $\gamma$. We demonstrated the effects of each hyperparameter in Section 4.3. Given the success of smoothing, it is desirable to find systematic ways to determine optimal hyperparameters based on an approximation of the underlying fitness landscape. We demonstrated our hyperparameter choices are not specific to either AAV or GFP, but this does not guarantee optimality for new landscapes. We believe the connections between spectral graph theory and protein optimization has more to give in advancing the important problem of protein optimization.

ACKNOWLEDGMENTS

The authors thank Hannes Stärk, Rachel Wu, Nathaniel Bennett, Sean Murphy, Jaedong Hwang, Josef Šivic, and Tomáš Pluskal for helpful discussion and feedback.

JY was supported in part by an NSF-GRFP. JY, RB, and TJ acknowledge support from NSF Expeditions grant (award 1918839: Collaborative Research: Understanding the World Through Code), Machine Learning for Pharmaceutical Discovery and Synthesis (MLPDS) consortium, the Abdul Latif Jameel Clinic for Machine Learning in Health, the DTRA Discovery of Medical Countermeasures Against New and Emerging (DOMANE) threats program, the DARPA Accelerated Molecular Discovery program and the Sanofi Computational Antibody Design grant. IF is supported by the Office of Naval Research, the Howard Hughes Medical Institute (HHMI), and NIH (NIMH-MH129046). RS was partly supported by the European Regional Development Fund under the project IMPACT (reg. no. CZ.02.1.01/0.0/0.0/15_003/0000468), the Ministry of Education, Youth and Sports of the Czech Republic through the e-INFRA CZ (ID:90254), and the MISTI Global Seed Funds under the MIT-Czech Republic Seed Fund.

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

## A    ADDITIONAL GFP ANALYSIS

**Design-bench difficulty.** Prior works have used the GFP task introduced by design-bench (DB), a suite of model-based reinforcement learning tasks (Trabucco et al., 2022), which samples a starting set of 5,000 sequences from the 50-60th percentile fitness range. However, we found this task to be too easy in the sense only one mutation was required from sequences in the training set to achieve the 99th percentile. We quantify this difficulty using the mutational gap described in eq. (6). Our proposed medium and hard difficulties (Section 4.1) require many more mutations to reach the top fitness percentile, see Figure 4. Similar issues may be present in other benchmarks.

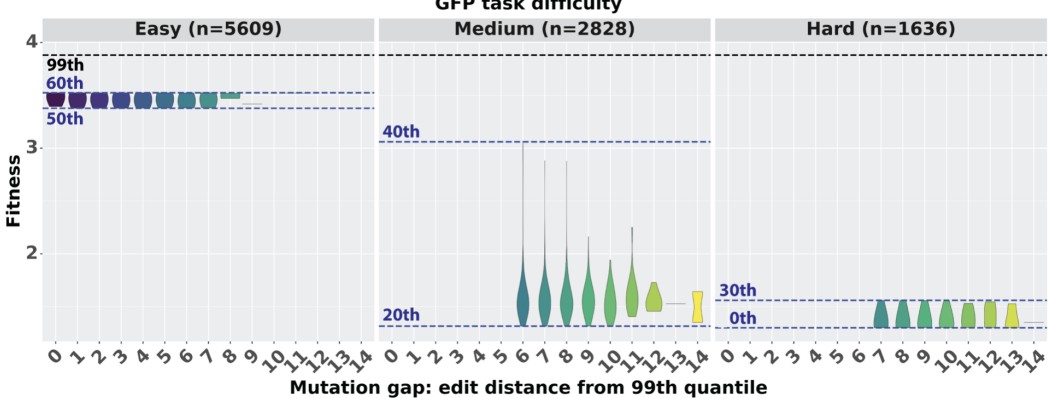

Figure 4: **Easy** is taken from design-bench where sequences between the 50-60th percentile are used in training regardless of edit distance to sequences in the 99th percentile. Data leakage is present due to multiple measurements that allows the wild-type and other top sequences to be included during training. **Medium** filters the training dataset to have sequences in the 20-40th percentile and be 6 or more mutations away from anything in the top 99th percentile. **Hard** similarly filters for sequences in at most the 30th percentile and 7 or more mutations away.

# B  ADDITIONAL METHODS

## B.1  CNN ARCHITECTURE

We utilize a 1D convolutional neural network (CNN) architecture in our model and oracle. The CNN takes in a one-hot encoded sequence as input then applies a 1D convolution with kernel width 5 followed by max-pooling and a dense layer to a single node that outputs a scalar value. It uses 256 channels throughout for a total of 157,000 parameters. Despite its simplicity, we find the CNN to outperform Transformers. Indeed, this corroborates the results in Dallago et al. (2021) that a simple CNN can be effective in low data regimes.

Training is performed with batch size 1024, ADAM optimizer (Kingma & Ba, 2014) (with $\beta_1 = 0.9, \beta_2 = 0.999$), learning rate 0.0001, and 50 epochs, using a single A6000 Nvidia GPU.

## B.2  METRICS

We provide mathematical definitions of each metric. Note $g_\phi$ is the evaluator trained to predict the approximate fitness as a proxy for experimental validation.

- **(Normalized) Fitness** = `median`$(\{\xi(\hat{x}_i; Y^*)\}_{i=1}^{N_\text{samples}})$ where $\xi(\hat{x}; Y^*) = \frac{g_\phi(\hat{x}_i) - \min(Y^*)}{\max(Y^*) - \min(Y^*)}$ is the min-max normalized fitness based on the lowest and highest known fitness in $Y^*$.

- **Diversity** = `median`$(\{\text{dist}(x, \tilde{x}) : x, \tilde{x} \in \hat{X}, x \neq \tilde{x}\})$ is the average sample similarity.

- **Novelty** = `median`$(\{\eta(\hat{x}_i; X)\}_{i=1}^{N_\text{samples}})$ where $\eta(x; X) = \min(\{\text{dist}(x, \tilde{x}) : \tilde{x} \in X^*, \tilde{x} \neq x\})$ is the minimum distance of sample $x$ to any of the starting sequences $X$.

---

**Algorithm 2** `Smooth`: Graph-based Smoothing

---

**Require:** Sequences: $X$
**Require:** Noisy model weights: $\tilde{\theta}$
 1: $V, E \leftarrow \texttt{CreateGraph}(X)$        ▷ Construct graph (Algorithm 4).
 2: $L \leftarrow \texttt{GraphLaplacian}(V, E)$        ▷ Compute graph Laplacian.
 3: $Y \leftarrow [f_{\tilde{\theta}}(x_1), \ldots, f_{\tilde{\theta}}(x_{N_\text{nodes}})]^\top$
 4: $\hat{Y} \leftarrow (\mathbb{I} + \gamma L)^{-1} Y$        ▷ Compute smoothed fitness labels.
 5: $\theta \leftarrow \arg\max_\theta \mathbb{E}_{(x,\hat{y}) \sim (V, \hat{Y})} [(\hat{y} - f_\theta(x))^2]$        ▷ Train on smoothed dataset.
 6: **Return** $\theta$

---

**Algorithm 3** `GWG`: Gibbs With Gradients

---

**Require:** Parent sequences: $X$
**Require:** Model weights: $\theta$
 1: $X' \leftarrow \emptyset$
 2: **for** $x \in X$ **do**
 3:     **for** $i = 1, \ldots, N_\text{prop}$ **do**        ▷ Number of proposals per sequence.
 4:         $x' \leftarrow x$
 5:         $(i^\text{loc}, j^\text{sub}) \sim q(\cdot|x)$        ▷ Sample index and token eq. (3)
 6:         $x'_{i^\text{loc}} \leftarrow \mathcal{V}_{j^\text{sub}}$        ▷ Apply mutation
 7:         **if** accept using eq. (4) **then**
 8:             $X' \leftarrow X' \cup \{x'\}$
 9:         **end if**
10:     **end for**
11: **end for**
12: **Return** $X'$        ▷ Return accepted sequences.

---

---

**Algorithm 4** `CreateGraph`

---

**Require:** Sequences: $X$
  1: $V \leftarrow X$                                                                ▷ Construct nodes.
  2: **while** $|V| \leq N_{\text{nodes}}$ **do**
  3:     $x \sim \mathcal{U}(V)$
  4:     $x' \leftarrow$ `PointMutation`$(x)$             ▷ Sample a point mutation uniformly at random.
  5: **end while**
  6: $E \leftarrow \bigcup_{x \in V}$ `kNN`$(x; V)$                      ▷ Construct edges (Algorithm 5).
  7: **Return** $(V, E)$

---

**Algorithm 5** `kNN`

---

**Require:** Current node: $x$
**Require:** All nodes: $V$
  1: $\mathcal{D}(x) \leftarrow \bigcup_{x' \in V / \{x\}} \text{dist}(x', x)$     ▷ Levenstein distance between every pair of sequences.
  2: $\mathcal{X}' \leftarrow \text{TopK}(\mathcal{D}(x), V)$                  ▷ Compute K closest sequences to $x$.
  3: $\text{E}(x) \leftarrow \bigcup_{x' \in \mathcal{X}'} (x, x')$               ▷ Construct neighborhood around $x$.
  4: **Return** $\text{E}(x)$

---

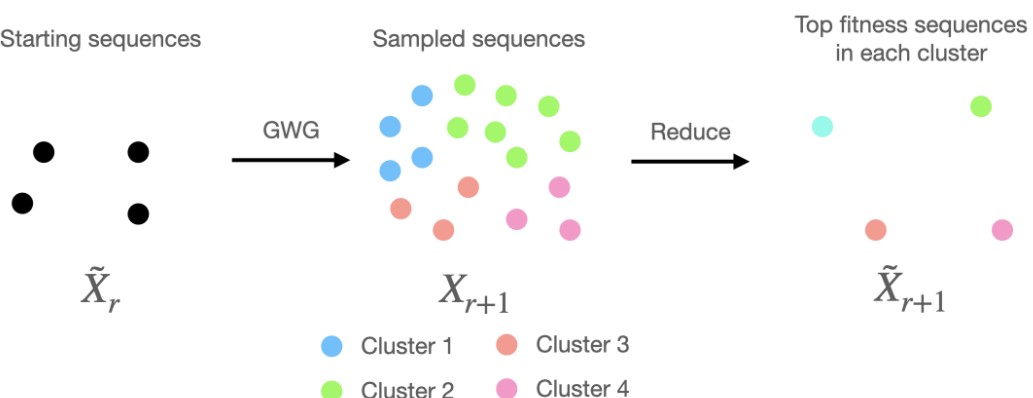

Figure 5: **Illustration of clustered sampling.** $\tilde{V}_r$ is the starting set of sequences for sampling in round $r$. GWG (Algorithm 3) is ran to generate many sample sequences, $V_{r+1}$. To control computation, we hierarchically cluster all sampled sequences based on Levenshtein distance and take the top fitness sequence in each cluster, using our trained fitness prediction model $f_\theta$ to score each sequence – we refer to this subroutine as `Reduce` (eq. (5)). The top sequences, $\tilde{V}_{r+1}$ are used for the next round.

## C   ADDITIONAL RESULTS

### C.1   SAMPLING TEMPERATURE SWEEP

We determine the effect of different tmperatures $\gamma$ when running GGS on the hard difficulty for GFP and AAV. All other hyperparameters follow those used in the main results, see Section 4.2. Table 5 shows the results where clearly $\gamma = 0.1$ performs the best for both AAV and GFP.

Table 5: Temperature sweep.

| Temperature ($\gamma$) | GFP hard | | | AAV hard | | |
|---|---|---|---|---|---|---|
| | Fitness | Diversity | Novelty | Fitness | Diversity | Novelty |
| 0.01 | 0.65 (0.0) | 5.3 (0.8) | 7.4 (0.5) | 0.45 (0.0) | 15.2 (1.1) | 9.0 (0.0) |
| 0.1 | 0.74 (0.0) | 3.6 (0.1) | 8.0 (0.0) | 0.6 (0.0) | 4.5 (0.2) | 7.0 (0.0) |
| 1.0 | 0.0 (0.1) | 28.2 (0.8) | 11.4 (0.5) | 0.45 (0.0) | 11.9 (0.5) | 8.0 (0.0) |
| 2.0 | 0.0 (0.1) | 36.1 (1.0) | 13.0 (0.0) | 0.33 (0.0) | 16.7 (0.9) | 8.5 (0.5) |

## C.2 SMOOTHING ANALYSIS

In this section, we provide further analyses into the effect of smoothing on performance of GGS, extrapolation to unseen data, and acceptance rate of the GWG sampling procedure. Throughout, we use the same parameters $\tau = 0.1, \gamma = 1, r = 15, N_{nodes} = 250,000$ as in the main text.

### C.2.1 ADDITIONAL BENCHMARKS

We first define additional benchmarks, one easier, and three harder, for each protein dataset.

Table 6: GFP extra tasks

| Difficulty | Range (%) | Gap | $|\mathcal{D}|$ |
|---|---|---|---|
| Easy | 50th-60th | 0 | 5609 |
| Harder1 | < 30th | 8 | 1129 |
| Harder2 | < 20th | 8 | 792 |
| Harder3 | < 10th | 8 | 397 |

Table 7: AAV extra tasks

| Difficulty | Range (%) | Gap | $|\mathcal{D}|$ |
|---|---|---|---|
| Easy | 50th-60th | 0 | 4413 |
| Harder1 | < 30th | 13 | 1157 |
| Harder2 | < 20th | 13 | 920 |
| Harder3 | < 10th | 13 | 476 |

We note that the "easy" GFP task is equivalent to the design-bench baseline that is sometimes used as a benchmark in protein engineering tasks. Due to experimental noise, protein variants are assayed multiple times, and can be assigned multiple fitness values, which means the fitness values of one sequence may occupy a large percentile *range*. In the case of this task, multiple measurements of the wildtype GFP fitness are found in the 50th-60th percentile range. Because WT GFP is also a "top sequence," this task necessarily has a mutational gap of 0. Due to this leakage, we develop our own benchmarks in the main text, and extend those to AAV.

### C.2.2 HOW SMOOTHING AFFECTS PERFORMANCE

The following two tables show how a smoothed model outperforms its unsmoothed counterpart according to our evaluator across all GFP/AAV benchmarks except AAV Harder2 (see ($*$)), and GFP Harder3, where the smoothing was not sufficient to induce successful GWG sampling (see Table 10).

Table 8: Smoothing improves GGS performance on GFP tasks

| Difficulty | Smoothed | Median Fitness | Diversity | Novelty |
|---|---|---|---|---|
| Easy | No | 0.05 | 24.83 | 13.36 |
| | Yes | **0.84** | 5.45 | 3.51 |
| Medium | No | 0.51 | 10.5 | 15.4 |
| | Yes | **0.76** | 3.7 | 5.0 |
| Hard | No | 0.10 | 23.02 | 16.8 |
| | Yes | **0.74** | 3.6 | 8.0 |
| Harder1 | No | 0.00 | 22.86 | 17.0 |
| | Yes | **0.67** | 4.45 | 9.12 |
| Harder2 | No | 0.00 | 22.22 | 16.5 |
| | Yes | **0.60** | 5.42 | 9.82 |
| Harder3 | No | 0.00 | 23.02 | 16.8 |
| | Yes | 0.00 | 15.73 | 21.2 |

For the GFP task, our model fails (achieves 0 median fitness) when we restrict the data to the 10th percentile and mutation gap 8 for GFP where $|\mathcal{D}| = 397$.

Table 9: Smoothing improves GGS performance on AAV tasks

| Difficulty | Smoothed | Median Fitness | Diversity | Novelty |
|---|---|---|---|---|
| Easy | No | 0.47 | 2.69 | 7.81 |
| | Yes | **0.49** | 9.18 | 7.99 |
| Medium | No | 0.37 | 6.60 | 6.62 |
| | Yes | **0.48** | 4.66 | 5.59 |
| Hard | No | 0.33 | 12.32 | 13.8 |
| | Yes | **0.60** | 4.5 | 7.0 |
| Harder1 | No | 0.30 | 0.53 | 6.00 |
| | Yes | **0.31** | 13.80 | 14.679 |
| Harder2 | No | **0.28***| 4.46 | 11.93 |
| | Yes | 0.27 | 15.98 | 19.41 |
| Harder3 | No | 0.25 | 3.08 | 5.63 |
| | Yes | **0.38** | 7.05 | 9.486 |

(*): The unsmoothed model only outperforms its smoothed counterpart when applying GWG to the unsmoothed model generates only a few unique sequences nearby to the starting set (as evidenced by the low novelty for this benchmark)

For AAV, we find the model is able to still find signal and achieve 0.384 evaluated fitness despite the data being limited to the 10th percentile and mutation gap of 13 where $|\mathcal{D}| = 476$. It is notable, though, that the performance improvements gained from smoothing are smaller than in the case of GFP. Presumably, this is due to the vastly reduced dimension of the AAV sequence space in comparison to that of GFP, which may result in a neural network to learn a smoother landscape without any regularization.

### C.2.3 HOW SMOOTHING AFFECTS EXTRAPOLATION + SAMPLING

The following tables show the benefits of smoothing on extrapolation to held out ground truth experimental data, up to a certain difficulty benchmark, as well as how smoothing vastly improves the acceptance rate for the GWG sampling procedure.

Table 10: Smoothing improves extrapolation and GWG sampling, up to GFP Harder3

| Difficulty | Smoothed | Train MAE | Holdout MAE | Acc. Rate |
|---|---|---|---|---|
| Easy | No | **0.03** | 0.99 | 0.02 |
| | Yes | 0.71 | **0.61** | **0.99** |
| Medium | No | **0.10** | 1.29 | 0.61 |
| | Yes | 0.20 | **0.88** | **0.62** |
| Hard | No | **0.06** | 1.44 | 0.01 |
| | Yes | 0.15 | **0.93** | **0.43** |
| Harder1 | No | **0.07** | 1.39 | 0.01 |
| | Yes | 0.15 | **0.94** | **0.43** |
| Harder2 | No | **0.01** | 1.41 | 0.01 |
| | Yes | 0.12 | **0.90** | **0.59** |
| Harder3 | No | 0.01 | **1.41** | 0.01 |
| | Yes | 0.01 | 1.42 | 0.01 |

Table 11: Smoothing improves extrapolation up to AAV Hard and GWG sampling on all AAV tasks

| Difficulty | Smoothed | Train MAE | Holdout MAE | Acc. Rate |
|---|---|---|---|---|
| Easy | No | **0.28** | 2.82 | 0.01 |
| | Yes | 1.76 | **2.28** | **0.99** |
| Medium | No | **0.35** | 3.12 | 0.01 |
| | Yes | 0.44 | **2.76** | **0.82** |
| Hard | No | **0.48** | 3.70 | 0.30 |
| | Yes | 0.55 | **3.09** | **0.78** |
| Harder1 | No | **0.66** | **3.99** | 0.01 |
| | Yes | 0.69 | 4.24 | **0.47** |
| Harder2 | No | **0.56** | **4.13** | 0.01 |
| | Yes | 0.58 | 4.37 | **0.55** |
| Harder3 | No | 0.47 | **4.58** | 0.01 |
| | Yes | 0.47 | 4.59 | **0.64** |

For each benchmark category, we evaluated the impact of smoothing on extrapolation abilities by analyzing the Mean Absolute Error (MAE) of the models on that benchmark's training and holdout datasets from the experimental ground truth. The effectiveness of smoothing was indicated by reduced MAE values on the holdout set. We also find that the MAE on the training set is lower for the unsmoothed models, as expected. In line with the results of the previous section, the effect of smoothing is reduced for AAV. As task difficulty increases, for both proteins, the effectiveness of smoothing on extrapolation decreases, which we expect as any signal leading from the training set to the fitter sequences gets obscured as training set size decreases.

Finally, we note that in every case except two, smoothing dramatically increases acceptance rate of the GWG sampling procedure, which aligns with the inversely proportional relationship between smoothness of the energy function and sampling efficiency. In the case of the hardest GFP task, even the the smoothed model had overfit to the training set. As for the GFP medium task, we suspect that this particular section of the experimental dataset allowed the unsmoothed model to learn a smooth landscape initially.

