# OpenReview forum: "Improving protein optimization with smoothed fitness landscapes"
_ICLR.cc/2024/Conference — ICLR 2024 poster_

### Official Review · Reviewer_zgX5 · 2023-10-25

**Soundness:** 3 good
**Presentation:** 2 fair
**Contribution:** 2 fair
**Rating:** 6
**Confidence:** 4

**Summary:**

The manuscript proposes a set of techniques for protein optimization. The first is a method for smoothing protein fitness landscapes. The second is a technique to optimizing in this landscape using the Gibbs With Gradients procedure, which has previously been shown to provide excellent results for discrete optimization. The authors also design two new optimization tasks based on the GFP and AAV datasets, which are designed to be more difficult than previous variants. Finally, the authors demonstrate empirically that their method performs competitively with the state-of-the-art.

**Strengths:**

### Originality
Although the GWG optimization procedure has been used in other contexts, the application to protein optimization is novel. To my knowledge, also the specific graph-based formulation of the regression problem itself is new.

### Quality
The paper seems technically sound. Code was provided to ensure reproducibility, and the authors provide additional details about the method in the supporting material.

### Significance
The paper does not give much insight into why the method outperform earlier approaches (see below for details), but the empirical results are convincing, which by itself could be sufficient to have impact on the growing subcommunity in ICML interested in protein modelling and design.

**Weaknesses:**

My main concern with the paper is that I - after reading it - do not feel much wiser about promising methodogical directions for protein modelling going forward.
What I lack in the paper is perhaps more of a motivation of why particular modelling choices were made. For instance, why is the Tikhunov regularization a meaningful choice in the context of protein optimization? Intuitively to me, it seems like a fairly crude choice, ignoring much of what we know about proteins already (e.g. that certain amino acids are biochemically similar to others). The paper also provides no biological intuition about why we would expect the smoothness would help. Presumably, the idea must be that there are different length scales to the problem, and that we can ignore the short length scales and focus on the longer ones - but it is not obvious to be why that would be the case for proteins. Is part of the explanation that experimental data is typically quite noisy? But if that's the case, you would assume that you would get similar behavior by using a simple GP with observation noise - just using a kernel based on edit distance - or based on Eucledian distance in one-hot space. The paper would be much more satisfying for me if the smoothing procedure was motivated more clearly, and perhaps even validated independent of the optimization procedure (I assume you would hope that the smoothed regressor would extrapolate better?)

My other serious concern is about the empirical evaluation of the model. As far as I can see, when we evaluate an optimization model against an oracle, there is a risk that we end up optimizing against extrapolation artefacts of the oracle, in particular if we end up evaluating it far away from the data it was trained on. My concern is whether your method has an unfair advantage compared to the baselines, because it uses the same CNN architecture for both the model and the oracle - and could therefore be particularly well suited for exploiting these artefacts. To rule out this concern, it would be interesting to see how the model performs against an oracle trained using a completely different model architecture.

**Questions:**

Page 4,
*"Edges, E, are constructed with a k-nearest neighbor graph around each node based on the Levenshtein distance 3."*
In real-world cases, the starting point is often a saturation mutagenesis experiment, where a lot of candidates will be generated with the same edit distance from the wild type (e.g. edit distance 1). In such cases, won’t the fixed k-out degree lead to an arbitrary graph structure (I mean, if the actual number of equidistant neighbors is much larger than k)?

Page 6, *"4.1 Benchmark"*
It was difficult to follow exactly what "develop a set of tasks" implies. Since the benchmarks are built from existing datasets, the authors should make it clearer exactly what they are "developing": is it only the starting set, or do they also restrict themselves to a subsample of the entire set? In table 1 and 2, are both *Range*, *|D|*, and *Gap* specifically selected for, or does e.g. *|D|* arise out of a constraint on *Range* and *Gap*?

Page 6. *"Oracle"*
Since you are using a CNN both as your oracle, and as the basis for your smoothed landscape model, isn’t there a risk that your model is just particularly well suited for exploiting the extrapolation artifacts of the oracle? (repetition of concern stated above).

### Minor comments:
Page 1, *"but high quality structures are not available in many cases"*
After AlphaFold, many would consider that high quality structures are now available in most cases.

Page 2, *"mutation is proposed renewed gradient computations."*
Something is wrong in this sentence

Page 3, *"in-silico oracles provides a accessible way for evaluation and is done in all prior works."*
This is not entirely accurate. People have optimized against actual experiments (e.g. Gruver, ..., Gordon-Wilson, 2023) - or optimized to find the optimal candidate in a fixed set of experimentally characterized proteins.

Page 4, eq (2) *"H(x)"*
As far as I can see, H(x) has not been introduced(?)

Page 6. *"we utilize a simpler CNN that achieves superior performance in terms of Spearman correlation and fewer false positives."*
Was this correlation measured on the GFP test set provided by TAPE after fitting on the training set?. If so, it's odd that the original TAPE paper did not find the CNN-based ResNet to outperform the transformer (actually, the transformer performance was dramatically higher). Please clarify.

Page 6. *"Recall the protein optimization task is to use D"*
Perhaps help the reader by rephrasing to "Recall the protein optimization task is to use the starting set D"

---

> ### Author Response · Authors · 2023-11-18
> **Response**
>
> We thank the reviewer for their time and detailed feedback. We appreciate the valid concerns and hope our response can address them. We have grouped together related points and itemized our response accordingly.
>
> ## Weaknesses
>
> > My main concern with the paper is that I - after reading it - do not feel much wiser about promising methodogical directions for protein modelling going forward. What I lack in the paper is perhaps more of a motivation of why particular modelling choices were made…The paper also provides no biological intuition about why we would expect the smoothness would help.
>
> The core takeaway of our work is to demonstrate the benefits of regularization through smoothing. We argue our modeling choices are based on protein knowledge. As we say in our introduction, “Proteins can be notorious for highly non-smooth fitness landscapes: fitness can change dramatically with single mutations, fitness measurements contain experimental noise, and most protein sequences have zero fitness.” It is well-known that deep learning models perform poorly with noisy and small datasets, i.e. curse of dimensionality. Knowing this, it becomes **crucial to remove the undesirable properties of protein data** and apply regularization if we want good performance with deep learning. For the purpose of protein engineering, **we care about finding the best sequences and not exactly modeling the true (un-smooth) fitness landscape**.
>
> > For instance, why is the Tikhunov regularization a meaningful choice in the context of protein optimization? Intuitively to me, it seems like a fairly crude choice, ignoring much of what we know about proteins already (e.g. that certain amino acids are biochemically similar to others).
>
> The purpose of Tikhunov regularization is to train a surrogate model of the fitness landscape that **is synergistic with DL-based sampling algorithms**. Our algorithm of choice is Gibbs With Gradients (GWG) which performs exceptionally well on (non-protein) discrete optimization benchmarks, but has not found success in protein sequence optimization. **The theory of GWG explicitly states the convergence of the Markov chain and its performance are related to the smoothness of the model**. We applied Tikunhov regularization to smooth the model and validated that GWG can achieve state-of-the-art performance if the correct regularization is applied.
>
> > Presumably, the idea must be that there are different length scales to the problem, and that we can ignore the short length scales and focus on the longer ones - but it is not obvious to be why that would be the case for proteins. Is part of the explanation that experimental data is typically quite noisy?
>
> Does short length scales refer to short mutational gaps (i.e. requiring few mutations to improve over the wild type)? If so, many protein engineering problems can require the need for long length scales. For instance, Somermeyer et al [1] shows multiple GFP proteins found in nature with cgreGFP having the highest fluorescence but only 41% sequence similarity with the avGFP used in our work. A priori it can be unknown how many mutations are needed to improve over the starting sequences. Furthermore, even for short length scales, the data can be limited and noisy which is where smoothing can still help – we see this in our results where most fail to get non-zero fitness using a un-smoothed model.
>
> > But if that's the case, you would assume that you would get similar behavior by using a simple GP with observation noise - just using a kernel based on edit distance - or based on Euclidean distance in one-hot space.
>
> One of our baselines, BOqei, is a GP. The GP was implemented as part of a baseline in CoMs [2] and included in follow-up works (e.g. [3]) where it was found to perform poorly especially for long proteins like GFP where it achieved the worst performance. GPs in general suffer from scaling to high-dimensional problems such as protein sequences [4].
>
> > The paper would be much more satisfying for me if the smoothing procedure was motivated more clearly,
>
> Thank you for the direct feedback! We hope our above answers have motivated smoothing. The introduction (third paragraph) mentions our motivation and Figure 1 provides geometric intuition of why smoothing helps with optimization. We have added a paragraph to the discussion about our modeling choices that depart from known fitness landscapes.

---

> ### Author Response · Authors · 2023-11-18
> **Response cont.**
>
> > and perhaps even validated independent of the optimization procedure (I assume you would hope that the smoothed regressor would extrapolate better?)
>
> We have included regression results that validate with smoothing in table 10 and 11. The results can be found in our global response. We took our unsmoothed and smoothed models trained on the hard and medium splits, and evaluated on the held-out data for which we have the true experimental fitness values for. We see that, as expected, smoothed models have higher training MAE but lower test MAE than their unsmoothed counterparts across all benchmarks, indicative of improved extrapolation as a result of smoothing.
>
> > My other serious concern is about the empirical evaluation of the model. As far as I can see, when we evaluate an optimization model against an oracle, there is a risk that we end up optimizing against extrapolation artefacts of the oracle, in particular if we end up evaluating it far away from the data it was trained on.
>
> This is a valid concern. However, we point out the prior protein engineering publications at ICLR, NeurIPS, and ICML have all used in-silico models to evaluate sequences. We have followed this practice. Some works have included experimental validation, but these authors have resources from biological collaborators or pharma companies [5].
>
> > My concern is whether your method has an unfair advantage compared to the baselines, because it uses the same CNN architecture for both the model and the oracle - and could therefore be particularly well suited for exploiting these artefacts. To rule out this concern, it would be interesting to see how the model performs against an oracle trained using a completely different model architecture.
> > Page 6. "Oracle" Since you are using a CNN both as your oracle, and as the basis for your smoothed landscape model, isn’t there a risk that your model is just particularly well suited for exploiting the extrapolation artifacts of the oracle? (repetition of concern stated above).
>
> We state in our work, “to ensure a fair comparison, we use the same model architecture in all baselines.” **We have ruled out the possibility of our method solely exploiting the model architecture.**
>
> ## Questions
>
> > Page 4, "Edges, E, are constructed with a k-nearest neighbor graph around each node based on the Levenshtein distance 3." In real-world cases, the starting point is often a saturation mutagenesis experiment, where a lot of candidates will be generated with the same edit distance from the wild type (e.g. edit distance 1). In such cases, won’t the fixed k-out degree lead to an arbitrary graph structure (I mean, if the actual number of equidistant neighbors is much larger than k)?
>
> Great point! A crucial component of our method is to augment the starting sequences by randomly mutating sequences to extrapolate the graph further than an edit distance of 1. With site saturation data, the starting sequences may have an arbitrary graph structure, but the more important sequences are the augmented ones that the model has not seen during training. These augmented sequences are >1 edit distance from the wild type and provide an approximation of how the model extrapolates. These predictions are smoothed if they are noise and we believe this procedure leads to improved sampling of better sequences.
>
> > Page 6, "4.1 Benchmark" It was difficult to follow exactly what "develop a set of tasks" implies. Since the benchmarks are built from existing datasets, the authors should make it clearer exactly what they are "developing": is it only the starting set, or do they also restrict themselves to a subsample of the entire set? In table 1 and 2, are both Range, |D|, and Gap specifically selected for, or does e.g. |D| arise out of a constraint on Range and Gap?
>
> We apologize for the confusion. Our benchmark trains the oracle on the entire dataset while the difficulties (which we call tasks) limit the starting set each method (including ours) has access to. |D| arises out of the choices for range and gap. Our choices for range and gap were chosen after experimenting with the values where we saw all the baselines failed on the hard starting set. We have changed the formatting of Table 1, placing the |D| column on the right, rather than the middle
>
> > Page 1, "but high quality structures are not available in many cases" After AlphaFold, many would consider that high quality structures are now available in most cases.
>
> While AlphaFold2 has made a revolution in the availability of structures, high quality structures are not readily available. The AlphaFold2 database paper [7] states only 58% of predicted residues have a confident prediction of which 36% have very high confidence. Furthermore, prediction of complexes lags behind the prediction of single chain. Lastly, an important point is AlphaFold2 has poor accuracy of point mutations [8].

---

> ### Author Response · Authors · 2023-11-18
> **Response cont. cont.**
>
> > Page 2, "mutation is proposed renewed gradient computations." Something is wrong in this sentence
>
> Thank you for this catch! We have corrected it.
>
> > Page 3, "in-silico oracles provides a accessible way for evaluation and is done in all prior works." This is not entirely accurate. People have optimized against actual experiments (e.g. Gruver, ..., Gordon-Wilson, 2023) - or optimized to find the optimal candidate in a fixed set of experimentally characterized proteins.
>
> Indeed, Gruver et al does perform in-vitro evaluation for antibodies. We have corrected the statement, “most prior works.” Note Gruver et al makes use of in-silico screening to first filter their designs before running in-vitro experiments.
>
> > Page 4, eq (2) "H(x)" As far as I can see, H(x) has not been introduced(?)
>
> We mention H(x) is the 1-Hamming ball in a foot note. We have now included a mathematical description in the text, "$H(x) = \{y \in V^M : d_(x, y) \leq 1 \}$ is the 1-ball around $x$ using Hamming distance."
>
> > Page 6. "we utilize a simpler CNN that achieves superior performance in terms of Spearman correlation and fewer false positives." Was this correlation measured on the GFP test set provided by TAPE after fitting on the training set?. If so, it's odd that the original TAPE paper did not find the CNN-based ResNet to outperform the transformer (actually, the transformer performance was dramatically higher). Please clarify.
>
> Yes, we evaluated the measurement on the held out set provided by TAPE. Note TAPE used a ResNet which has far more parameters and is a more complicated model than the simple 1D CNN used in our work and in [6] where they observed a similar finding of the CNN outperforming TAPE.
>
> > Page 6. "Recall the protein optimization task is to use D" Perhaps help the reader by rephrasing to "Recall the protein optimization task is to use the starting set D"
>
> Thank you for this improvement. We have applied the suggestion.
>
> [1] https://elifesciences.org/articles/75842.pdf
>
> [2] https://arxiv.org/pdf/2202.08450.pdf
>
> [3] https://arxiv.org/abs/2203.04115
>
> [4] https://arxiv.org/abs/1810.12283
>
> [5] https://arxiv.org/abs/2203.12742
>
> [6] https://www.biorxiv.org/content/10.1101/2021.11.09.467890v1
>
> [7] https://www.nature.com/articles/s41586-021-03828-1
>
> [8] https://www.ncbi.nlm.nih.gov/pmc/articles/PMC10019719

---

> > ### Comment · Reviewer_zgX5 · 2023-11-22
> > **Response to rebuttal**
> >
> > Thanks to the authors for their detailed rebuttal. The new table with regression results is a welcome addition, and I appreciate the authors attempt to motivate the smoothing for me.
> >
> > Also thanks to the reviewers for pointing out that the same model architecture was used in all baselines. I had missed that. I still think that there is a potential issue here, since you might primarily be testing the impact of smoothing on the artefacts of the model, rather than the actual biological fitness landscape. But at least it helps that you are doing the same for all baselines.
> >
> > I'll increase my score to a 6.

---

> > > ### Author Response · Authors · 2023-11-23
> > > **Response (and thanks)**
> > >
> > > Thank you for the discussion and raise in score! As a final point before the discussion period ends, we would like to respond to the potential issue raised as there may be a potential confusion. To be clear, there are two models being used.
> > >
> > > - Fitness prediction model, $f_\theta$, that only sees a subset of the entire data (to simulate starting with a limited set). This model is smoothed and used in the baselines and ours.
> > > - Evaluator model, $g_\phi$, that is **trained on the entire dataset and is not smoothed**. This is the model we evaluate our sequences with and none of the methods have access to. Note this is the best in-silico proxy we can produce for the real biological fitness landscape.
> > >
> > > As you can see, **we find using a smoothed model $f_\theta$ can generalize to improved performance on a non-smoothed model $g_\phi$**, in other words our closest proxy for fitness landscape. This is evidence to show smoothing can help generalize.
> > >
> > > Additionally, **our regression results show the smoothed model can generalize to potentially unsmoothed data**. The unsmoothed model has better training loss, thus modeling the fitness landscape more accurately, but has poor generalization on the held out set where it really matters to sample better sequences.
> > >
> > > We hope this addresses the potential issue and help motivate the contributions in our work. Unfortunately the only other argument is to run real experimental validation but this is costly and not in our capabilities at the time. Moreover, very few papers at ICLR include experimental validation. Regardless, thank you very much for the interaction and time!
> > >
> > > (Please excuse our deleted message where we accidentally hit post before finishing)

---

> > > > ### Comment · Reviewer_zgX5 · 2023-11-23
> > > > **Response to clarification**
> > > >
> > > > Yes. I understand that you train two models. But as far as I can see, in the optimization setting, you will apply them both in an out-of-domain setting by pushing them outside the space of observed mutations. It is not clear to me that the smoothness properties of the real biological fitness landscape are preserved even by the non-smoothed model out-of-domain, and therefore it is unclear whether the improvements you see would translate to a real scenario.
> > > >
> > > > What others have done is to optimize against models trained across the entire space of proteins (e.g. stability) - which presumably would make such 'oracles' more robust in the extrapolation setting. But I completely agree that it is difficult to assess this completely without running the actual experiments.
> > > >
> > > > I do acknowledge that your regression experiments demonstrate that the smoothing has a positive effect in-domain, so thanks for including those.

---

### Official Review · Reviewer_7UDm · 2023-10-31

**Soundness:** 3 good
**Presentation:** 3 good
**Contribution:** 3 good
**Rating:** 6
**Confidence:** 3

**Summary:**

The authors introduces a method called Gibbs sampling with Graph-based Smoothing (GGS) that uses Tikunov regularization and graph signals to smooth the protein fitness landscape, improving the ability to create diverse, functional sequences.

**Strengths:**

Figure 1 is very helpful in the understanding of this approach.

My understanding of the section described in Section 3.2 is relatively clear.

I think the Fitness, Diversity, and Novelty scores to be interpretable and helpful.

I think it is encouraging that graph-based smoothing (GS) helps almost all other methods in Table 3. It’s also great that this is a relatively straightforward procedure.

**Weaknesses:**

“While dWJS is an alternative approach to fitness regularization, it was only demonstrated for antibody optimization. To the best of our knowledge, we are the first to apply discrete regularization using graph-based smoothing techniques for general protein optimization.” - This doesn’t seem justifiably novel. Proteins are proteins.

Generally, I wouldn’t use the term “fitness” when describing protein function. Rather, I would use phenotype or function, as fitness is a broad, poorly defined subset of fitness.

Figure 5 is a reason why these function predictors should not be called “oracles”, because mapping the effect of mutation to function is difficult itself. I’d prefer “protein function approximator”, or something along those lines.

“These were chosen due to their long lengths, 237 and 28 residues” What do you mean here? 28 isn’t that long. I realize it is in the context of a larger protein, but I’d be clear about that.

**Questions:**

For the smoothing procedure, it’d be great to show the amount of error introduced into the labels of the sequences. For instances where either a reasonable oracle model exists, or sequences with large hamming distances have been measured, and this smoothing procedure is introduced, what is the correlation of function values before and after?

“To control compute bandwidth, we perform hierarchical clustering (Mullner, 2011) on all the se- ¨ quences in a round and take the sequence of each cluster with the highest predicted fitness using fθ.” Why not use the “noisy model” for this, because it is the oracle for the true fitness of a sequence?

“Section 4.1 presents a set of challenging tasks based on the GFP and AAV proteins that emulate starting optimization with a noisy and limited set of proteins.” I would like the authors to be clear by what the mean by “noisy”. Is it experimental noise? Is the landscape too sparsely sampled? Where is this noise coming from, and what relative distribution does it have?

Generally, I feel like Figure 5 is a distraction from the broader utility of the work. I’d just cite Dallago 2021 like you did for the use of CNNs.

---

> ### Author Response · Authors · 2023-11-18
> **Response**
>
> We thank the reviewer for their time and feedback. Below we address the issues and comments.
>
> ## Weaknesses
>
> > “While dWJS is an alternative approach to fitness regularization, it was only demonstrated for antibody optimization. To the best of our knowledge, we are the first to apply discrete regularization using graph-based smoothing techniques for general protein optimization.” - This doesn’t seem justifiably novel. Proteins are proteins.”
>
> We apologize for the poor choice in wording. dWJS is a concurrent work that performs smoothing by training their energy-based model with denoising. We meant to highlight the differences in smoothing approaches where we instead use graph-based smoothing in protein optimization. We have replaced the line with, “dWJS trains by denoising to smooth a energy-based model whereas we apply discrete regularization using graph-based smoothing techniques.”
>
> > Generally, I wouldn’t use the term “fitness” when describing protein function. Rather, I would use phenotype or function, as fitness is a broad, poorly defined subset of fitness.
>
> We internally have debated over the right terminology. We noticed the protein engineering field has settled on using fitness landscape rather than function landscape [1,2,3]. We chose to stick with fitness to follow this convention. Our very first sentence mentions “fitness can be defined as performance on a desired property or function.” We would prefer to keep the wording to avoid major changes, but are happy to change all occurrences if the reviewer views this as a major issue.
>
> > Figure 5 is a reason why these function predictors should not be called “oracles”, because mapping the effect of mutation to function is difficult itself. I’d prefer “protein function approximator”, or something along those lines.
>
> Thank you for the suggestion! We have changed the mention of oracle everywhere to “trained evaluator model.” We agree the term oracle places too much trust in the in-silico evaluations.
>
> > “These were chosen due to their long lengths, 237 and 28 residues” What do you mean here? 28 isn’t that long. I realize it is in the context of a larger protein, but I’d be clear about that.
>
> Thank you for pointing this out. We have changed this sentence to highlight the data availability, “these were chosen due to their relatively large amount of measurements, 56,806 and 44,156 respectively, with sequence variability of up to 15 mutations from the wild-type.”
>
> ## Questions
>
> > For the smoothing procedure, it’d be great to show the amount of error introduced into the labels of the sequences. For instances where either a reasonable oracle model exists, or sequences with large hamming distances have been measured, and this smoothing procedure is introduced, what is the correlation of function values before and after?
>
> We have included error analysis of the train and test splits for predicting the true fitness. See the global response. We see there is an improvement in generalization error to the held out test set with smoothing. The effect is greater with GFP than AAV.
>
> > “To control compute bandwidth, we perform hierarchical clustering (Mullner, 2011) on all the se- ¨ quences in a round and take the sequence of each cluster with the highest predicted fitness using fθ.” Why not use the “noisy model” for this, because it is the oracle for the true fitness of a sequence?
>
> We do not use the noisy model, denoted $f_{\tilde{\theta}}$, because the predictions from the noisy model are inaccurate (cf. above table). We use the smoothed model $f_{\tilde{\theta}}$ instead for higher quality predictions. We find using the noisy model for our method leads to very poor performance (2nd to last row in tables 3 and 4).
>
> > “Section 4.1 presents a set of challenging tasks based on the GFP and AAV proteins that emulate starting optimization with a noisy and limited set of proteins.” I would like the authors to be clear by what the mean by “noisy”. Is it experimental noise? Is the landscape too sparsely sampled? Where is this noise coming from, and what relative distribution does it have?
>
> This is a great point! By noisy, we mean the experimental noise. We briefly mention this in the introduction, “fitness measurements contain experimental noise.” We have clarified the sentence to say, “Section 4.1 presents a set of challenging tasks based on the GFP and AAV proteins that emulate starting with experimental noise and a sparsely sampled fitness landscape.”
>
> > Generally, I feel like Figure 5 is a distraction from the broader utility of the work. I’d just cite Dallago 2021 like you did for the use of CNNs.
>
> Thank you for the suggestion. We have cited Dallago and removed Figure 5.
>
> [1] https://www.nature.com/articles/nrm2805
>
> [2] https://www.biorxiv.org/content/10.1101/2021.11.09.467890v1
>
> [3] https://www.nature.com/articles/s41592-019-0496-6

---

### Official Review · Reviewer_Ayis · 2023-10-31

**Soundness:** 3 good
**Presentation:** 3 good
**Contribution:** 3 good
**Rating:** 6
**Confidence:** 3

**Summary:**

This study proposes to smooth the protein fitness landscape to facilitate protein fitness optimization using gradient based techniques. This is motivated by the ruggedness of protein fitness landscape which makes optimization challenging. A graph based smoothing technique for fitness landscape followed by Gibbs with Gradient sampling is used to perform protein fitness optimization. Evaluation of their method has been done on train sets designed from GFP and AAV with two degrees of difficulty defined by the mutational gap between the starting set and the optimum in the dataset (not included in the starting set).  Their method shows better performance than others in the proposed benchmark. The proposed graph smoothing technique has been shown to help with other methods as well.

**Strengths:**

Designing train sets with varying difficulties for the task of optimization.
Proposing a new method for smoothing the protein fitness landscape before optimization.

**Weaknesses:**

The proposed method has many hyperparameters to tune.
Given certain properties of protein fitness landscape, smoothing can hurt if not done properly.

**Questions:**

1)	Please explain why after smoothing, the diversity and novelty of the final set of sequences decreases.
2)	In defining train sets with varying levels of difficulty only two medium (mutation gap 6) and hard (mutation gap 7) levels have been used. What happens if you make this harder (higher than 7)? Also, should we assume that for less mutational gap all methods perform comparably?
3)	As stated in the paper, single mutations can dramatically change the fitness. In the smoothing performed, similar sequences are enforced to have similar fitness. Have you investigated where smoothing can be detrimental?
4)	How is the number of proposals ($N_{\text{prop}}$) per sequence set?
5)	Have you tried smaller sizes for the starting set? In real world problems the size of the starting set could be much smaller than 2000?
6)	Was the oracle only used at the end for performance evaluation? In AdaLead, did you use the oracle as the fitness landscape or $f_\theta$?
7)	Mention the augmented graph size (how does it change with the size of the sequence)
8)	Minor: In Eq 5, $X_0$ should be X.

---

> ### Author Response · Authors · 2023-11-18
> **Response**
>
> We thank the reviewer for their time and feedback. Below we address the issues and comments.
>
> ## Weaknesses
>
> > The proposed method has many hyperparameters to tune. Given certain properties of protein fitness landscape, smoothing can hurt if not done properly.
>
> We require four method-specific hyperparameters that we analyze in Section 4.3 and Appendix C.1 as well as providing sweeps for each of them. (We don’t consider CNN model and training hyperparameters since we use standard choices from prior works.) **Please note that other works require equal or more method-specific hyperparameters than ours: COMs requires six, DynaPPO requires 4, GFlowNets require 4 hyperparameters**. Our work is forthcoming about the hyperparameters and provides analysis into each of them:
>
> 1. *Sampling temperature* $\tau$. Appendix C.1 provides a table of sweeping over different temperatures. Temperature is not specific to us since every MCMC method needs to tune the temperature for the respective need of balancing the sample quality and diversity.
>
> 2. *GWG round* $R$. Again the number of rounds is not specific to our method but is required for GWG. Figure 3 shows $R$ needs to be high enough for the Markov chain to mix and reach its stationary distribution.
>
> 3. *Number of nodes in the protein graph* $N_{nodes}$ is a hyperparameter specific to our method. Figure 3 shows higher $N_{nodes}$ to give better performance, but higher values lead to slower run time. Our choice of $N_{nodes}=250,000$ results in our method taking < 1 hour to sample and still achieves state-of-the-art result.
>
> 4. *Smoothness weight* $\gamma$ is the other hyperparameter specific to our method. Figure 3 shows the choice $\gamma$ is important for the final performance as too high can lead to lower performance. **We explicitly state in Section 5 that determining $\gamma$ is a current limitation of our method and automatic ways of choosing $\gamma$ is a important future direction.** Choosing $\gamma$ is non-trivial since protein landscapes are very difficult. Nevertheless, our work demonstrates the importance of careful smoothing is in protein optimization.
>
> ## Questions
>
> > 1. Please explain why after smoothing, the diversity and novelty of the final set of sequences decreases.
>
> Our metrics reflect the tradeoff between exploitation and exploration where a method needs to maximize the reward (fitness in our case) while discovering the multiple modes of the distribution. Reward and diversity are often at odds since many diverse sequences can be sampled but each have low reward (and vice versa).
> Tables 3 and 4 show this general trend in methods without smoothing where higher diversity and novelty leads to the worse fitness (see GFN-AL and BOqei). Smoothing will reduce diversity by flattening the local and noisy variation in the fitness landscape (see the unsmoothed and smoothed landscapes in Figure 1). Our results show smoothing is effective to increase fitness across methods while avoiding low quality sequences that could artificially increase diversity.

---

> > ### Author Response · Authors · 2023-11-18
> > **Response cont.**
> >
> > > 2. In defining train sets with varying levels of difficulty only two medium (mutation gap 6) and hard (mutation gap 7) levels have been used. What happens if you make this harder (higher than 7)? Also, should we assume that for less mutational gap all methods perform comparably?
> >
> > We perform extensive study in more mutation gap and starting range settings. Our results are in Appendix C.2 and summarized here.
> > We use a consistent mutation gap of 8 in the case of GFP, and 13 in the case of AAV, while decreasing the range percentile cutoffs. We do this to control how much smaller the starting dataset is getting as we increase the difficulty, as increasing the mutation gap can sharply decrease $|D|$.
> >
> > In the manuscript, we stopped at the hard difficulty, since other baselines fail at this point. We find that for GFP, if we restrict the range to the bottom 10 percent of sequences and enforce a mutational gap of at least 8, then our smoothing method fails. However, for AAV, we find the model is able to still find signal and achieve 0.384 fitness despite the data being limited to the 10th percentile and mutation gap of 13 where $|D|=476$. Presumably, the difference arises as a result of the different contours of the fitness landscapes of the AAV and GFP
> >
> > Finally, although we were not able to reproduce the results of an easy task for all baselines during the rebuttal period, we would agree with your assumption that as the mutational gap decreases, all methods should perform comparably.
> >
> > **GFP Alternative Benchmarks (Mutation Gap = 8)**
> >
> > | Range | D | Smoothed | Acceptance Rate | Median Fitness | Diversity | Novelty |
> > |-------|---|----------|-----------------|----------------|-----------|---------|
> > | 0-30% | 1129 | No | 0.01 | 0.00 | 22.86 | 17.0 |
> > |       |      | Yes | 0.43 | 0.67 | 4.45 | 9.12 |
> > | 0-20% | 792  | No | 0.01 | 0.00 | 22.22 | 16.48 |
> > |       |      | Yes | 0.59 | 0.60 | 5.42 | 9.82 |
> > | 0-10% | 397  | No | 0.01 | 0.00 | 23.02 | 16.82 |
> > |       |      | Yes | 0.01 | 0.00 | 15.73 | 21.17 |
> >
> > **AAV Smaller Benchmarks (Mutation Gap = 13)**
> >
> > | Range | D | Smoothed | Acceptance Rate | Median Fitness | Diversity | Novelty |
> > |-------|---|----------|-----------------|----------------|-----------|---------|
> > | 0-30% | 1157 | No | 0.001 | 0.30 | 0.53 | 6 |
> > |       |      | Yes | 0.473 | 0.31 | 13.80 | 14.69 |
> > | 0-20% | 920  | No | 0.001 | 0.39 | 6.43 | 9.13 |
> > |       |      | Yes | 0.546 | 0.27 | 15.98 | 19.41 |
> > | 0-10% | 476  | No | 0.001 | 0.25 | 3.08 | 5.63 |
> > |       |      | Yes | 0.64 | 0.38 | 7.05 | 9.48 |
> >
> >
> > > 3. As stated in the paper, single mutations can dramatically change the fitness. In the smoothing performed, similar sequences are enforced to have similar fitness. Have you investigated where smoothing can be detrimental?
> >
> > Yes, we have explored different amounts of smoothing in Figure 3 where we show setting the smoothing weight $\gamma$ too high can be detrimental. Clearly too much smoothing destroys the signal and is detrimental. We note setting the optimal $\gamma$ as a limitation of our method in Section 5 (Discussion) and is an important future direction that requires non-trivial theoretical results. Our work demonstrates the importance of smoothing that other works have overlooked.
> >
> > > 4. How is the number of proposals (N_{seq}) per sequence set?
> >
> > We set $N_{seq}$ to a reasonable value to keep run time low for our method since clustering can be expensive; our method takes 20 minutes time with $N_{seq}=100$ on a A6000 GPU. Higher would result in better results or faster convergence (i.e. requiring lower number of rounds). Therefore $N_{seq}$ is compute dependent.
> >
> > > 5. Have you tried smaller sizes for the starting set? In real world problems the size of the starting set could be much smaller than 2000?
> >
> > See the response to Q2 which analyzes the effect of decreasing dataset size. **Note that our starting sets are far below what other methods have utilized. I.e. design-bench starts off with 5000 sequences in their benchmarks.**
> >
> > > 6. Was the oracle only used at the end for performance evaluation? In AdaLead, did you use the oracle as the fitness landscape or ?
> >
> > Yes, the oracle was only used at the end for performance evaluation. Adalead did not use the oracle as the fitness landscape. We provide Adalead with the smoothed and unsmoothed fitness function $f_\theta$ that we use for our method. The same $f_\theta$ is used across all our baselines when possible to allow for a fair comparison.
> >
> > > 7. Mention the augmented graph size (how does it change with the size of the sequence)
> >
> > We mention the augmented graph size in Section 4 “we augment the graph until it has Nnodes = 250 000 nodes.” This is used regardless of sequence length; we use the same graph size for both GFP and AAV.
> >
> > > 8. Minor: In Eq 5, X_0 should be X.
> >
> > Thank you for this catch!

---

### Official Review · Reviewer_PzPY · 2023-11-05

**Soundness:** 1 poor
**Presentation:** 1 poor
**Contribution:** 2 fair
**Rating:** 3
**Confidence:** 3

**Summary:**

The paper propose a smoothing method on fitness function given a protein sequence. Assume that the given original data set is small, authors proposed a sampling augmentation method and a TV smoothing regulariser. After which MCMC algorithm is use to further optimise the fit.

**Strengths:**

Authors presented some good results on benchmark datasets.

**Weaknesses:**

The paper is hard to read and understand. I itemise areas for improvements.

1. Having one figure to show overall flow of logic could help. Fig1 seems to do the job. There are some confusion between training and sampling. I understand that the author first train f(x) and then use f(x) as a surrogate function for MCMC optimisation. This point does not come out naturally.

2. construction of KNN graph could be described more clearly. (see Eq above Eq(1))

3. Symbols of Eq.(2) are ill defined. The authors should provide in the appendix some details of GWG and reference the appendix in the main text.

4. Eq.(4) should give the acceptance rate. while q are the probability of trial moves. x and x' are two states for jumping in this one MC step. Usual notation is q(x|x') vs q(x'|x), notation of Eq.(4) certainly is not of this form. Instead i^loc and j^sub and being used. The same i^loc and j^sub cannot appear in both numerator and denominator of Eq.(4).

5. Eq.(4) what is the temperature of this move? It seems the temperature is set to 1. Why is the temperature 1? Is there any annealing process?

6. Clustered sampling section should be explained better.

**Questions:**

Is there a way to test that the surrogate function by itself is good enough? The authors look at the overall performance that could infer to correctness of the surrogate function.


see above section on 'weakness'

---

> ### Author Response · Authors · 2023-11-18
> **Response**
>
> We thank the reviewer for their time and feedback. We notice the reviewer gave low scores for Soundedness, Presentation, and Contribution. However, we are unable to find reasoning behind the poor Soundedness and Contribution.
> We kindly ask if the reviewer can elaborate on their low scores to help improve our work and address them in the discussion period.
> We believe the clarity has improved with the reviewer’s insightful comments. See our responses below.
>
> ## Weaknesses and questions
>
> > The paper is hard to read and understand. I itemise areas for improvements.
>
> We thank the reviewer for pointing out issues regarding clarity.
>
> > 1. Having one figure to show overall flow of logic could help. Fig1 seems to do the job. There are some confusion between training and sampling. I understand that the author first train f(x) and then use f(x) as a surrogate function for MCMC optimisation. This point does not come out naturally.
>
> Indeed, Figure 1 is meant to convey the overall flow of logic. By using the same notation, $f_\theta$, in the training (top row) and sampling (bottom row), we hoped it explains training and inference with $f_\theta$. We have updated figure 1 to clearly indicate that sampling follows after training and the same model $f_\theta$ is used in both training and sampling.
>
> > 2. construction of KNN graph could be described more clearly. (see Eq above Eq(1))
>
> Thank you for the suggestion. We have added a line near Eq (1) to state: “The graph construction algorithm can be found in Algorithm 4.” Our graph construction pseudocode used to be in Algorithm 2 but we have placed it in its own Algorithm now for clarity.
> We are unsure what the reviewer meant by “(see Eq above Eq(1))” since this equation refers to the definition of Total variation.
>
> > 3. Symbols of Eq.(2) are ill defined. The authors should provide in the appendix some details of GWG and reference the appendix in the main text.
>
> We apologize for confusion and typos. We have clarified in the text that we use the one-hot representation, have rewritten Eq (2) to explicitly describe the matrix algebra, and made sure every symbol is described. Please see our updated writing in Section 3.3.
> We believe section 3.3 fully describes the GWG algorithm as introduced in Grathwohl et al. Note Eq (4), (5), and (6) in Grathwohl et al are given as Eq (2), (3) in our work while Algorithm 3 in the appendix describes the full GWG algorithm (Algorithm 1 in Grathwohl et al). Our original submission states, “We provide the GWG algorithm in Algorithm 3.” We have added in Section 3.3, “In this section, we describe the GWG procedure for protein sequences.” We are happy to answer any specific concerns or questions the reviewer has regarding GWG.
>
> > 4. Eq.(4) should give the acceptance rate. while q are the probability of trial moves. x and x' are two states for jumping in this one MC step.
>
> The acceptance rate differs based on the learned energy function which is non-trivial to analyze.
> We are not sure what the reviewer means by “Eq. (4) should give the acceptance rate” when there is no theoretical method to derive the acceptance rate when $f_\theta$ is a neural network (to the best of our knowledge).
> However, we have added in the appendix tables 10 and 11 which includes a column containing the acceptance rates for each task and difficulty including whether we use our smoothed method or not.
> We note that the acceptance rate is higher for the smoothed version of our method as further evidence smoothing is necessary for GWG to work.
>
> **GFP**
> | Difficulty | Smoothed | Acc. Rate |
> |------------|----------|-----------|
> | Medium     | No       | 0.61      |
> |            | Yes      | **0.62**      |
> | Hard       | No       | 0.01      |
> |            | Yes      | **0.43**      |
>
> **AAV**
> | Difficulty | Smoothed | Acc. Rate |
> |------------|----------|-----------|
> | Medium     | No       | 0.01      |
> |            | Yes      | **0.82**      |
> | Hard       | No       | 0.30      |
> |            | Yes      | **0.78**      |
>
> > Usual notation is q(x|x') vs q(x'|x), notation of Eq.(4) certainly is not of this form. Instead i^loc and j^sub and being used. The same i^loc and j^sub cannot appear in both numerator and denominator of Eq.(4).
>
> Thank you for this catch! This was a mistake and we have corrected the MH-step with q(x|x') vs q(x'|x).

---

> > ### Author Response · Authors · 2023-11-18
> > **Response cont.**
> >
> > > 5. Eq.(4) what is the temperature of this move? It seems the temperature is set to 1. Why is the temperature 1? Is there any annealing process?
> >
> > The temperature is given in equation (3) where we state “$\tau$ is the sampling temperature.” The temperature is not 1. Our initial submission provided a sweep over different temperatures in table 5 (replicated below) and our best method uses $\tau=0.1$. Note it is common to sweep over temperatures and it is application specific of trading off diversity and sample quality.
> > | Temperature ($\gamma$) | GFP hard - Fitness | GFP hard - Diversity | GFP hard - Novelty | AAV hard - Fitness | AAV hard - Diversity | AAV hard - Novelty |
> > |------------------------|--------------------|----------------------|--------------------|--------------------|----------------------|--------------------|
> > | 0.01                   | 0.65 (0.0)         | 5.3 (0.8)            | 7.4 (0.5)          | 0.45 (0.0)         | 15.2 (1.1)           | 9.0 (0.0)          |
> > | 0.1                    | **0.74** (0.0)         | 3.6 (0.1)            | 8.0 (0.0)          | **0.6** (0.0)          | 4.5 (0.2)            | 7.0 (0.0)          |
> > | 1.0                    | 0.0 (0.1)          | 28.2 (0.8)           | 11.4 (0.5)         | 0.45 (0.0)         | 11.9 (0.5)           | 8.0 (0.0)          |
> > | 2.0                    | 0.0 (0.1)          | 36.1 (1.0)           | 13.0 (0.0)         | 0.33 (0.0)         | 16.7 (0.9)           | 8.5 (0.5)          |
> >
> > > 6. Clustered sampling section should be explained better.
> >
> > We apologize for lack of clarity with the clustered sampling. Due to limited space, we have added an additional figure in the appendix that illustrates clustered sampling. Section 3.3 now states “An illustration of clustering is provided in Figure 6.”
> > We are happy to address any specic component of clustered sampling that was not clear.
> >
> > > Is there a way to test that the surrogate function by itself is good enough? The authors look at the overall performance that could infer to correctness of the surrogate function.
> >
> > See our global response. We have analyzed the overall prediction error of our surrogate function with and without smoothing in each of our settings and found smoothness to be necessary in achieving improved generalization to unseen sequences.

---

> > > ### Comment · Reviewer_PzPY · 2023-11-22
> > > **keep score**
> > >
> > > I like to keep my score. I can see efforts in improving the paper. if the author work on the paper again for future submission, it should become good enough.

---

> > > > ### Author Response · Authors · 2023-11-22
> > > > **Request for justification**
> > > >
> > > > Once again we're grateful for the time the reviewer has spent providing feedback. **However, we believe we have addressed every point made in your review**.
> > > > It would help us understand what weaknesses and issues remain, **especifically regarding the poor Soundness and Contribution scores since there is no reasoning for these scores**.
> > > > We would greatly appreciate if the reviewer can provide justification and specify what needs improvement.

---

### Author Response · Authors · 2023-11-18
**Global response**

We thank the reviewers for their constructive feedback and time. Reviewers noted the novelty of our work in the difficulty benchmarks (reviewer Ayis), graph-based formulation of fitness regression (reviewer zgX5), and the general improvements of smoothing across multiple methods (reviewer 7UDm). In the global response, we highlight important changes to our manuscript while specific points are addressed in our reviewer responses.

## Smoothing analysis

We have added tables 10 and 11 that elucidates the effect of smoothing the fitness prediction model $f_\theta$. We analyze the Mean Absolute Error (MAE) of the train and test sequences of each splits where the test set are all the sequences and fitness pairs not used during training. We find the unsmoothed model has lower train MAE on all splits but higher test MAE than the smoothed model. This provides additional evidence of why smoothing helps during sampling: on top of smoother gradients, the generalization error is far lower.

**GFP task**

| Difficulty | Smoothed | Train MAE | Held-out MAE |
|------------|----------|-----------|--------------|
| Medium     | No       | **0.35**      | 3.18         |
| Medium     | Yes      | 0.45      | **2.76**         |
| Hard       | No       | **0.48**      | 3.70         |
| Hard       | Yes      | 0.56      | **3.09**         |

**AAV task**

| Difficulty | Smoothed | Train MAE | Held-out MAE |
|------------|----------|-----------|--------------|
| Medium     | No       | **0.11**      | 1.29         |
| Medium     | Yes      | 0.2       | **0.88**         |
| Hard       | No       | **0.06**      | 1.44         |
| Hard       | Yes      | 0.15      | **0.92**         |

**Additional difficulties.** We have added additional difficulties in tables 8 and 9 to analyze our method, GGS, beyond the hard difficulty. Our baselines all fail (without smoothing) on hard while GGS could still find improved sequences. We explored limiting the training set further to the 10th fitness percentile and <500 sequences with higher mutational gap if necessary. We find in these very low data regimes that GGS performance degrades and fails on GFP but still manages to discover signal in AAV. This points to the robustness of smoothing.

## Improved writing
Several reviewers pointed out issues with clarity around our sampling method and terminology, i.e. incorrect use of “oracle”. We have corrected any confusing statements and added explanations in red in the updated manuscript.

---

### Meta-Review · Area_Chair_fz1W · 2023-12-05

**Metareview:**

The manuscript introduces Gibbs sampling with Graph-based Smoothing (GGS), a novel approach for protein optimization, aiming to revolutionize biotechnology and medicine. Utilizing the well known graph Tikunov/Tikhonov regularization, GGS smooths the protein fitness landscape, enabling more effective optimization in a traditionally challenging, rugged landscape. This method, leveraging discrete energy-based models and Markov Chain Monte Carlo (MCMC) techniques, demonstrates superior performance in GFP and AAV benchmarks, achieving a 2.5-fold fitness improvement over its training set. The study addresses the limitations of small datasets through a sampling augmentation method and TV smoothing regularizer. Overall, GGS shows potential for optimizing proteins in limited data scenarios, representing a significant advancement in protein design.

**Justification For Why Not Higher Score:**

The manuscript, while introducing an innovative approach in protein optimization, exhibits several key limitations. Firstly, it lacks a comparison to the dWJS method, which is a conceptually similar in approach, leaving a gap in the evaluation of its relative effectiveness. Secondly, the reliance on a learned oracle for validation limits insights into the actual performance of the method, as it does not accurately reflect real-world scenarios. Lastly, the smoothing technique employed could be enhanced by incorporating considerations of the properties and embeddings of proteins, which would potentially improve the accuracy and applicability of the results. These limitations suggest areas for further work.

**Justification For Why Not Lower Score:**

The strengths of the manuscript are centered around the novelty and simplicity of its approach. The concept introduced is innovative in the field of protein optimization. Furthermore, this approach demonstrates enhanced performance across various applications, indicating its broad utility. The effectiveness of the method has been successfully validated in silico, providing a robust foundation for its practical applicability. Additionally, new results presented during the rebuttal period shed light on the positive impact of smoothing the fitness prediction model, further reinforcing the method's effectiveness. These strengths collectively highlight the potential of this approach in advancing protein optimization research.

---

### Decision · Program_Chairs · 2024-01-16

Accept (poster)